# Purinergic adipocyte-macrophage crosstalk promotes degeneration of thermogenic brown adipose tissue

Michelle Y Jaeckstein[1], Alexander W Fischer[1], Björn Rissiek [2], Tobias Staehler [3], Markus Heine[1], Janina Behrens[1], Oliver Mann[4], Alexander Pfeifer [5], Tim Magnus[2], Christian Schlein[6], Anna Worthmann[1], Ludger Scheja[1], Friedrich Koch-Nolte [3] & Joerg Heeren [1✉]

## Abstract

**Loss of brown adipose tissue (BAT) activity observed during ageing, obesity and living at thermoneutrality is associated with lipid accumulation, fibrosis and tissue inflammation in BAT. The mechanisms that promote this degenerative process of BAT remain largely enigmatic. Here, we show that an imbalance between sympathetic activation and mitochondrial energy handling causes BAT degeneration, which leads to impaired energy expenditure and systemic metabolic disturbances. Mechanistically, we demonstrate that brown adipocytes secrete ATP in response to imbalanced thermogenic activation, which activates P2X4 and P2X7 of BAT-resident macrophages. Notably, mice lacking activity of these purinergic receptors in myeloid cells are protected against BAT inflammation, thermogenic dysfunction and systemic metabolic disturbances under conditions of imbalanced BAT activation, thermoneutrality or overnutrition. These results highlight the relevance of extracellular ATP released by brown adipocytes as a paracrine signal for myeloid cells to initiate BAT degeneration.**

**Keywords** Adaptive Thermogenesis; Dyslipidemia; Hyperglycemia; Inflammation; P2X Receptors
**Subject Category** Metabolism

## Introduction

Brown adipose tissue (BAT) produces heat in response to cold exposure, and its presence and activity correlate with improved metabolic health, lower body weight and beneficial cardiovascular outcomes in humans (Becher et al, 2021; Saito et al, 2009; van Marken Lichtenbelt et al, 2009). The activity of BAT declines with ageing and is reduced in obese states both in rodents and in humans (Cypess et al, 2009; Heine et al, 2018; Roberts-Toler et al, 2015; Saito et al, 2009; van Marken Lichtenbelt et al, 2009). Accordingly, activating BAT or preventing the obesity- and ageing-induced decrease in BAT function have been proposed as a strategy to combat cardiometabolic diseases (Chondronikola et al, 2014; Chondronikola et al, 2016; Sakers et al, 2022; Scheja and Heeren, 2019; Yoneshiro et al, 2013). BAT-dependent energy expenditure is primarily triggered by norepinephrine that is released from sympathetic nerve endings, activating β-adrenergic receptors (Blondin et al, 2020; Cero et al, 2021). The associated adrenergic downstream signaling pathway stimulates intracellular lipolysis, and the released fatty acids are utilized for energy combustion through mitochondrial β-oxidation (Cannon and Nedergaard, 2004). At the same time, fatty acids activate uncoupling protein 1 (UCP1), a protein localized in the inner mitochondrial membrane of thermogenic adipocytes that generates heat by separating the proton gradient from ATP production (Cannon and Nedergaard, 2004). The sympathetic tone in BAT, and thus the activity of brown adipocytes, is very low at thermoneutrality, a condition in which the basal metabolic rate is sufficient to maintain euthermia of mammals. Thermoneutral conditions for mice are at an ambient temperature between 28 °C and 30 °C, whereas for lean humans the thermoneutral zone depends on various factors such as clothing and ranges from 22 °C to 30 °C (Fischer et al, 2018; Kingma et al, 2012). Notably, the latter represents a mild cold stimulus for mice that is associated with an elevated sympathetic tone and thermogenic activation of BAT. Accordingly, housing mice at 30 °C is closer to typical human energy metabolism (Fischer et al, 2018), and increases the propensity for developing obesity-associated comorbidities with characteristic phenotypes more closely resembling human inflammatory metabolic diseases (Ganeshan and Chawla, 2017; Giles et al, 2017; Seeley and MacDougald, 2021). In this context, it is important to note that a low sympathetic tone caused by thermoneutral housing conditions or by surgical denervation promotes BAT degeneration. This degenerative

[1]Department of Biochemistry and Molecular Cell Biology, University Medical Center Hamburg-Eppendorf, Hamburg, Germany. [2]Department of Neurology, University Medical Center Hamburg-Eppendorf, Hamburg, Germany. [3]Institute of Immunology, University Medical Center Hamburg-Eppendorf, Hamburg, Germany. [4]Department of General, Visceral and Thoracic Surgery, University Medical Center Hamburg-Eppendorf, Hamburg, Germany. [5]Institute of Pharmacology and Toxicology, University Hospital, University of Bonn, Bonn, Germany. [6]Institute of Human Genetics, University Medical Center Hamburg-Eppendorf, Hamburg, Germany. ✉E-mail: heeren@uke.de

process results in extensive lipid accumulation, loss of mitochondria and thermogenic proteins such as UCP1, activation of inflammatory pathways and a higher abundance of pro-inflammatory immune cells including macrophages (de Jong et al, 2019; Fischer et al, 2020b; Kotzbeck et al, 2018; Schlein et al, 2021). Notably, a similar phenotype has been described for UCP1-deficient mice exposed to cold (Bond et al, 2018; Kazak et al, 2017). In response to adrenergic signaling, brown adipocytes of these transgenic mice liberate fatty acids that are, however, not efficiently oxidized by the UCP1-deficient mitochondria. This creates a condition of energetic imbalance that can be considered as futile thermogenic activation causing BAT inflammation and degeneration.

The molecular and cellular mechanisms that initiate and mediate inflammatory tissue remodeling during BAT degeneration remain to be identified. Moreover, it is unclear whether potential intervention strategies targeting inflammation-related degeneration of BAT could be used to maintain functional capacity of thermogenic adipocytes. In the present study, we show by brown adipose tissue denervation that sympathetic tone is essential for inflammation and immune cell infiltration in BAT of UCP1-deficient mice. Using a pharmacological model mimicking futile thermogenic activation, we identify purinergic signaling through the ATP-activated ion channels P2X4 and P2X7 as the initial trigger of BAT degeneration. The combined inhibition of these purinergic receptors protects against BAT inflammation and dysfunction elicited by futile thermogenic activation or by prolonged exposure to thermoneutrality. Taken together, these results emphasize the relevance of a paracrine purinergic signaling axis for BAT degeneration. Moreover, this degenerative tissue-specific mechanism can be targeted pharmacologically, offering the opportunity to counteract the development of metabolic disturbances by maintaining BAT function.

# Results

## Sympathetic innervation is critical for BAT inflammation in UCP1-deficient mice

At housing temperatures below thermoneutrality, BAT receives a constant stimulatory input from the sympathetic nervous system (Cannon and Nedergaard, 2004). In BAT of $Ucp1^{-/-}$ mice, this cannot translate into full thermogenic activation due to the lack of UCP1-mediated non-shivering thermogenesis. This results in higher BAT weights that can be explained by higher lipid deposition upon whitening of the tissue (Fischer et al, 2020a). Another important phenotype of these mice is inflammation of BAT that is observed during cold-induced activation but not under thermoneutral housing conditions (Bond et al, 2018; Kazak et al, 2017). In line, we found significantly higher BAT weights, increased expression of pro-inflammatory and fibrosis marker genes, while body weights were unaffected in BAT of $Ucp1^{-/-}$ mice housed at 22 °C (Fig. EV1A–C). In the same setup, we detected an increased number of macrophages in BAT, which was accompanied by profound fibrosis (Figs. 1A and EV1D–G). Lowering the housing temperature to 6 °C to further increase the sympathetic tone and thermogenic activation (Cannon and Nedergaard, 2004) caused even higher inflammation as suggested by strongly elevated MAC2

protein levels in BAT depots of $Ucp1^{-/-}$ but not wild-type mice (Fig. EV1H,I). To directly assess the role of sympathetic activity for inflammation in BAT of UCP1-deficient mice, we denervated BAT of $Ucp1^{-/-}$ and control mice. As expected, the surgical procedure resulted in lower expression of $Ucp1$ in wild-type mice (Fig. 1B,C), and a virtual absence of the sympathetic innervation marker tyrosine hydroxylase and norepinephrine levels in BAT of both genotypes (Figs. 1C,D and EV1J). In one sample, full denervation was not achieved but was apparently clearly above 50% inhibition and did not change the interpretation of the results. In wild-type mice, denervation caused pro-inflammatory remodeling of BAT as indicated by increased expression of cytokines, immune cell and fibrosis markers (Fig. 1E). Notably, MAC2 protein expression, the inflammatory and fibrotic phenotype of $Ucp1^{-/-}$ mice was prevented by BAT denervation (Figs. 1C,E and EV1J). These data indicate that UCP1 deficiency in combination with sympathetic activation causes an imbalanced thermogenic response triggering severe BAT inflammation and fibrosis.

## Imbalanced BAT activation causes tissue degeneration and impaired thermogenesis

To mechanistically study inflammatory cell recruitment and tissue damage resulting from imbalanced thermogenic activation, we established a pharmacological in vivo model that combines adrenergic BAT stimulation with the inhibition of mitochondrial fatty acid oxidation, which together mimics unproductive thermogenesis in UCP1-deficient brown adipocytes. For this purpose, we used etomoxir (Eto), a compound inhibiting the mitochondrial fatty acid transport protein carnitine palmitoyltransferase 1 in combination with the sympathomimetic β3-adrenergic agonist CL316,243 (CL). The combined treatment with etomoxir and CL but not the administration of the individual compounds for three consecutive days resulted in lipid accumulation, fibrosis, higher abundance of MAC2-positive macrophages (Figs. 1F and EV2A–D) and lower $Ucp1$ expression (Fig. 1G). Consistently, gene expression of inflammatory, immune cell and fibrosis markers was strongly increased in BAT of Eto/CL-treated mice (Fig. 1H), indicating profound inflammation and BAT degeneration. This effect on inflammation was tissue-specific, as it was not observed in other metabolically active tissues such as inguinal white adipose tissue and liver (Figs. 1I and EV2E). As expected (Grujic et al, 1997; Himms-Hagen et al, 1994), CL treatment significantly increased energy expenditure in control mice (Fig. 1J,K). Notably, the combined Eto/CL treatment regimen was associated with reduced energy expenditure and lower body core temperature after CL injection assessed by indirect calorimetry (Fig. 1J–L). The inhibitory effect of etomoxir on mitochondrial lipid utilization was demonstrated by the higher respiratory quotient (Fig. EV2F,G), which as a readout for substrate utilization indicates higher glucose and reduced fatty acid oxidation. Taken together, the pharmacological treatment regimen based on etomoxir and CL caused inflammatory BAT degeneration.

## BAT degeneration is preceded by the infiltration of pro-inflammatory myeloid cells

Various signs of end stage BAT degeneration including infiltration of different immune cells and fibrosis were detected after three days

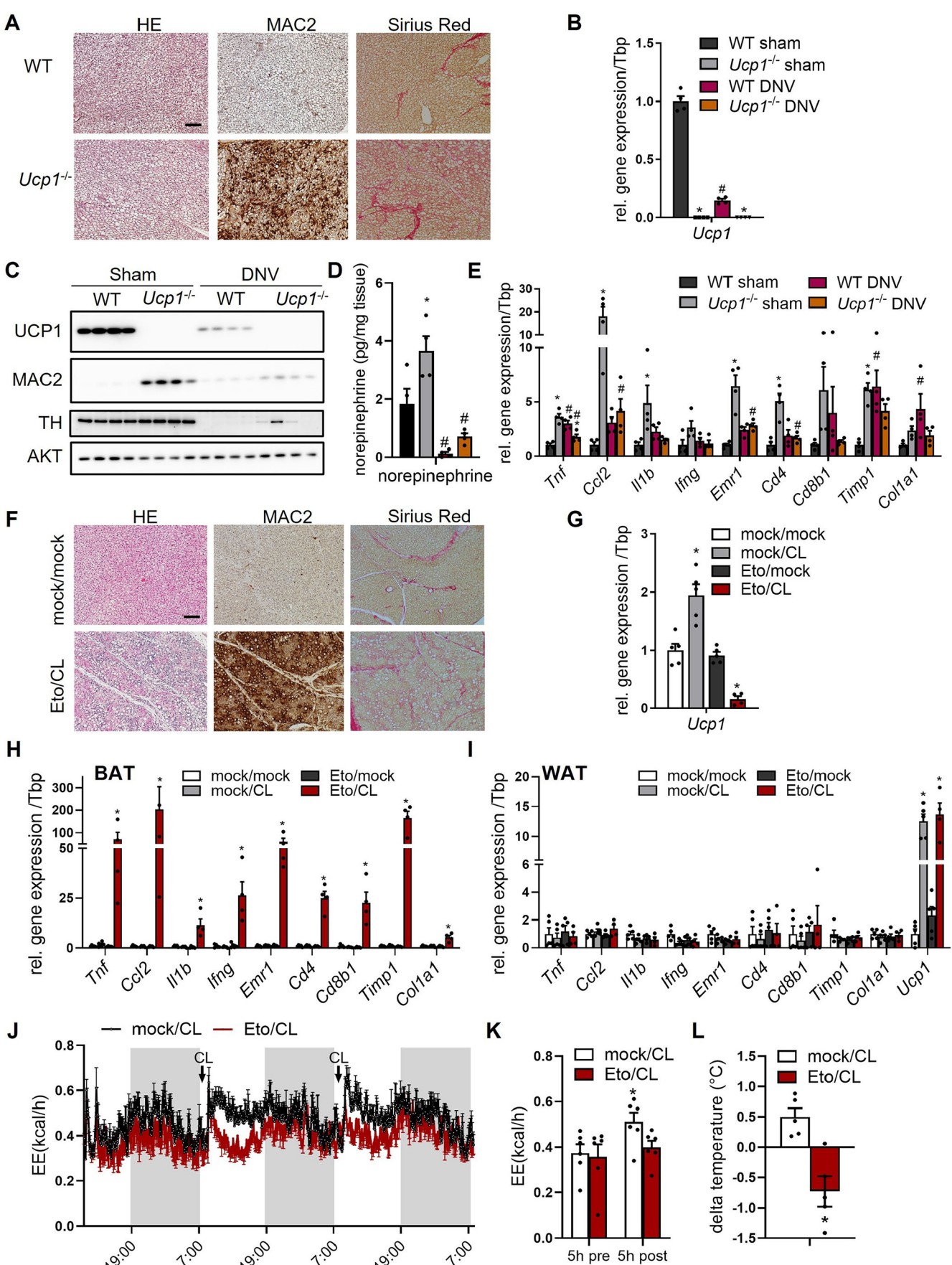

◄    **Figure 1.  Inflammatory BAT remodeling is triggered by imbalanced BAT activation.**

(A–E) Male $Ucp1^{-/-}$ mice and wild-type (WT) littermates were housed at 22 °C. (A) Representative BAT images of HE and Sirius Red stainings as well as immunostaining of MAC2. Scale bar, 50 µm. (B) $Ucp1$ expression in BAT of denervated (DNV) and sham-operated (sham) mice ($n = 4$). (*$P = <0.0001$, #$P = <0.0001$, *$P = 0.0055$; left to right). (C) Western Blot of BAT of DNV and sham mice. AKT served as loading control ($n = 4$). (D) Norepinephrine levels in BAT of DNV and sham $Ucp1^{-/-}$ and WT mice ($n = 4$). (*$P = 0.0212$, #$P = 0.0303$, #$P = 0.0006$; left to right). (E) Gene expression of inflammatory markers in BAT of DNV and sham mice ($n = 4$). ($Tnf$: *$P = <0.0001$, *$P = 0.0007$, *$P = 0.0255$; #$P = 0.0012$; $Ccl2$: *$P = 0.0006$, #$P = 0.0029$; $Il1b$: *$P = 0.0369$; $Emr1$: *$P = <0.0001$, #$P = 0.0020$; $Cd4$: *$P = 0.0004$, #$P = 0.0015$; $Timp1$: *$P = 0.0071$, #$P = 0.0049$; $Col1a1$: #$P = 0.0445$; left to right). (F–I) Wild-type mice housed at 22 °C were injected daily for three subsequent days with either vehicle (mock/mock), with the β3-adrenergic agonist CL316,243 alone (mock/CL), with the inhibitor of β-oxidation etomoxir alone (Eto/mock) or with the combination of both (Eto/CL). (F) Representative images of BAT from mock/mock and Eto/CL mice employing HE, Sirius Red and MAC2 (immune)-stainings, respectively. Scale bar, 50 µm. (G) $Ucp1$ gene expression in BAT ($n = 4$–5). (*$P = 0.0002$, *$P = 0.0009$; left to right). (H) Gene expression of inflammatory marker genes in BAT ($n = 4$–5). ($Tnf$: *$P = 0.0042$; $Ccl2$: *$P = 0.0128$; $Il1b$: *$P = <0.0001$; $Ifng$: *$P = <0.0001$; $Emr1$: *$P = <0.0001$; $Cd4$: *$P = <0.0001$; $Cd8b1$: *$P = <0.0001$; $Timp1$: *$P = <0.0001$; $Col1a1$: *$P = <0.0001$; left to right). (I) Gene expression of inflammatory marker genes in inguinal WAT ($n = 4$–5). ($Tnf$: *$P = <0.0001$, *$P = <0.0001$; left to right). (J) Energy expenditure (EE) in mock/CL- and Eto/CL-treated mice. The first and second CL injection are indicated by arrows ($n = 6$). (K) Quantification of energy expenditure 5 h pre and post first CL injection ($n = 6$). (*$P = <0.0001$). (L) Changes in body core temperature in response to CL injection ($n = 6$). (*$P = 0.0026$). Data are presented as mean values ± SEM. *$P < 0.05$ by two-way ANOVA comparing WT vs. $Ucp1^{-/-}$ (B, D, E), by ANOVA compared to mock/mock (G–I), by ANOVA comparing all groups (K) or Student's $t$ test (L). #$P < 0.05$ by two-way ANOVA comparing sham vs DNV (B, D, E). $N$ values indicate biological replicates. See also related Figs. EV1 and EV2. Source data are available online for this figure.

of combined Eto/CL treatment. As we were primarily interested in the pathways initiating the degenerative process, we investigated earlier time points of Eto/CL-induced BAT degeneration. To characterize the dynamics of BAT inflammatory remodeling, we performed a kinetic experiment where we analyzed BAT after the combined treatment with etomoxir and CL for 4, 8, 24 and 48 h, respectively. Already after 4 h, we observed a profound increase in the expression of the chemoattractant factor $Ccl2$ (Fig. 2A). This preceded the induction of pro-inflammatory cytokines ($Tnfa$, $Il1b$) and the myeloid cell marker $Emr1$, which were massively increased after 8 h (Fig. 2B). Expression of these genes remained at very high levels during the course of degeneration, while higher expression of $Cd4$ and $Cd8b1$ indicating T cell infiltration could only be detected 48 h after Eto/CL treatment (Fig. 2A–D). Notably, the reduction of $Ucp1$ expression in response to the treatment with etomoxir and CL is only observed at later time points and is preceded by extensive inflammatory response and lipolytic activation at earlier time points (Figs. 2A–D and EV2H,I). In line with gene expression data, immune cell phenotyping using flow cytometry revealed that infiltration of myeloid cells including neutrophils and macrophage subsets characterized the early phase of BAT degeneration (Fig. 2E–H). Other immune cells such as NK cells, CD4- and CD8-positive T lymphocytes and B cells do not change in numbers at the beginning but rose at 48 h only (Fig. 2I–L). Overall, the kinetic data provide evidence that pro-inflammatory myeloid cells contribute to the initiation of the degenerative process.

## Brown adipocyte dysfunction causes ATP secretion and is linked to purinergic receptor signaling in BAT

To delineate signaling pathways triggering BAT degeneration in response to imbalanced thermogenic activation, we performed bulk RNAseq analysis of BAT from mice treated with Eto/CL and compared expression to vehicle controls. Analysis of differentially expressed genes showed significant regulation of nearly 10,000 genes in response to imbalanced thermogenic activation (Fig. 2M; Dataset EV1), suggesting profound cellular and tissue remodeling. Consistent with the results presented in Fig. 1, $Ucp1$ expression was reduced, whereas non-canonical thermogenic genes such as $Atp2a2$, $Ckmt2$ and $Alpl$ were unaltered (Dataset EV1). Gene ontology (GO) enrichment analysis revealed that various pathways involved in

immune regulation were among the most represented biological processes (Fig. 2N; Dataset EV1). Remarkably, a highly significant enrichment was also observed for purine nucleotide and ATP metabolism. Overall, these data indicated that inflammatory pathways regulated by purinergic signaling could be activated under these conditions of BAT degeneration. Previously, it was described that etomoxir inhibits the polarization in macrophages and suppresses the differentiation of thermogenic adipocytes due to off-targets in vitro (Divakaruni et al, 2018; Shimura et al, 2025). This was based on lower expression levels of IL4-induced genes, e.g., $Retnla$, $Mgl2$ and $Ym1$, and of thermogenic genes such as $Dio2$ and $Ucp1$ (Divakaruni et al, 2018; Shimura et al, 2025). Both processes could affect the validity of our brown adipose tissue (BAT) degeneration model. However, RNA-seq data clearly show that Eto treatment alone did not affect the content of macrophages or immune cells, macrophage polarization or the expression of thermogenic genes in BAT in vivo (see Dataset EV1).

Based on these results, we quantified candidate pro-inflammatory molecules including cytokines, free fatty acids (FFA) and ATP in the supernatants of primary brown adipocytes treated with etomoxir, CL or the combination of these compounds (Fig. 3A–D). Secreted levels of the cytokines CCL2 and IL1β were largely unaltered by the treatments (Fig. 3A,B). In line, gene expression of pro-inflammatory cytokines was unchanged or even lower after Eto/CL treatment (Fig. EV3A). FFA were increased in the culture media of CL-treated brown adipocytes (Fig. 3C), which is consistent with efficient stimulation of intracellular triglyceride lipolysis through β3-adrenergic signaling. Notably, the combination of etomoxir and CL but not the single treatments resulted in the accumulation of extracellular ATP (Fig. 3D). In line, the imbalance of energy expenditure and lipolysis also lead to higher concentration of both FFA and ATP in the supernatants of UCP1-deficient primary brown adipocytes (Fig. 3E,F). To define the underlying pathway of ATP secretion, we evaluated RNAseq data and found elevated expression of the ATP channel pannexin 1 but lower levels for pannexin 2 (Fig. 3G,H). Based on the in vivo data, we blocked pannexin 1 channel activity in primary brown adipocytes and found blunted Eto/CL-induced ATP secretion (Fig. 3I). This indicates that ATP is released by stressed adipocytes in an active manner and is not a result of cell damage. As extracellular ATP is a pro-inflammatory factor, these findings prompted us to screen for

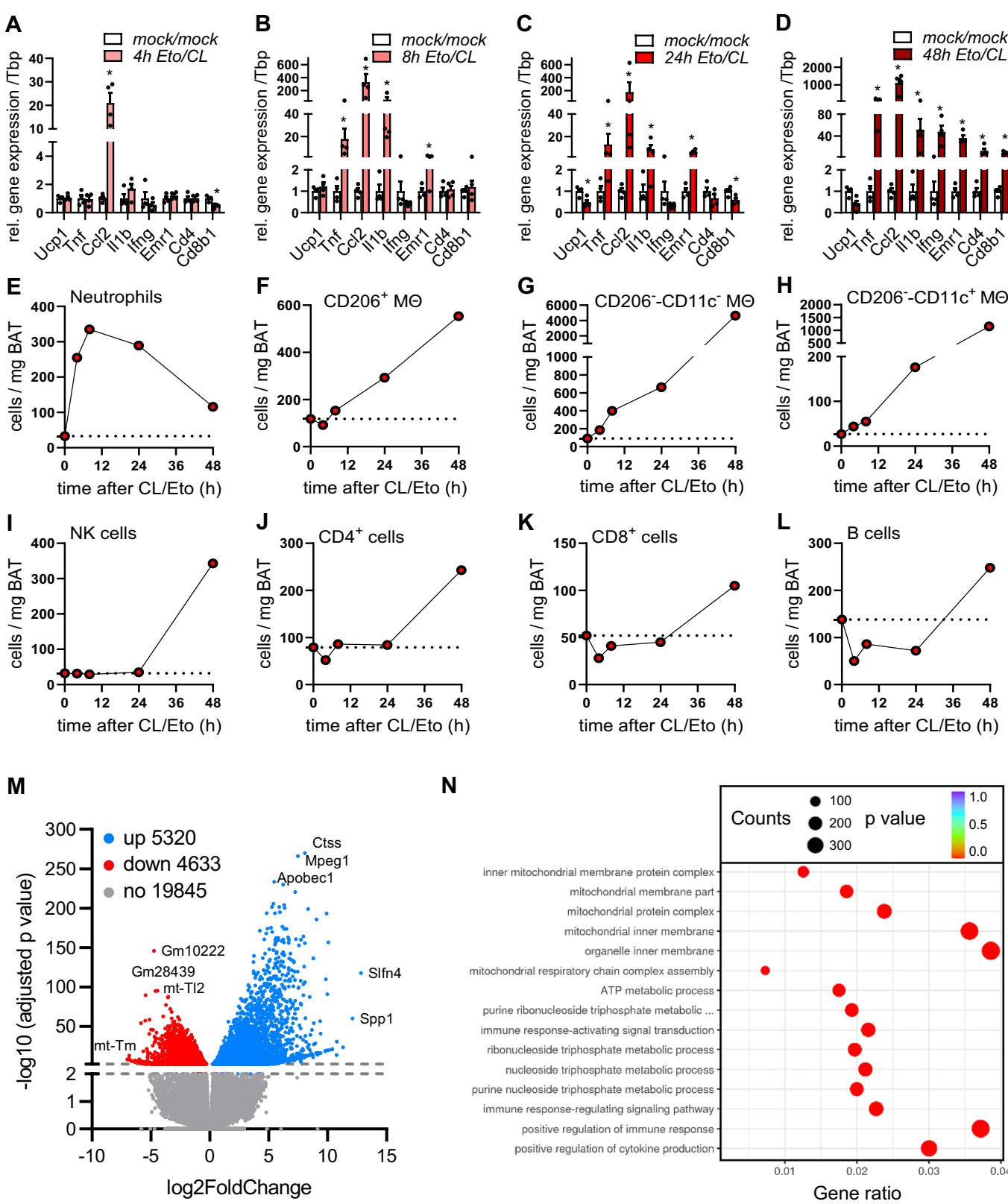

Figure 2. BAT degeneration is preceded by the infiltration of pro-inflammatory myeloid cells.

(A–L) Wild-type mice housed at 22 °C were injected for indicated time spans or (M, N) daily for three consecutive days with either vehicle (mock/mock) or with the combination of the β3-adrenergic agonist CL316,243 and the inhibitor of β-oxidation etomoxir (Eto/CL). (A) Gene expression of *Ucp1* and inflammatory marker genes in BAT; 4 h after Eto/CL injection (n = 4). (*Ccl2*: *P = 2.65E-05; *Cd8b1*: *P = 0.0103; left to right). (B) Gene expression of *Ucp1* and inflammatory marker genes in BAT; 8 h after Eto/CL injection (n = 4). (*Tnf*: *P = 0.0025; *Ccl2*: *P = 0.000016; *Il1b*: *P = 0.0005; *Emr1*: *P = 0.0442; left to right). (C) Gene expression of *Ucp1* and inflammatory marker genes in BAT; 24 h after Eto/CL injection (n = 4). (*Ucp1*: *P = 0.0199; *Tnf*: *P = 0.0410; *Ccl2*: *P = 0.0049; *Il1b*: *P = 0.0245; *Emr1*: *P = 0.0001; *Cd8b1*: *P = 0.0332; left to right). (D) Gene expression of *Ucp1* and inflammatory marker genes in BAT; 48 h after Eto/CL injection (n = 4). (*Tnf*: *P = 2.14E-05; *Ccl2*: *P = 1.73E-06; *Il1b*: *P = 0.0008; *Ifng*: *P = 0.0001; *Emr1*: *P = 1.79E-06; *Cd4*: *P = 0.0003; *Cd8b1*: *P = 3.76E-05; left to right). (E) Quantification of neutrophils in BAT determined by FACS analysis over the time span of 0–48 h after Eto/CL treatment. (F) Quantification of CD206+ MΘ cells in BAT determined by FACS analysis over the time span of 0–48 h after Eto/CL treatment. (G) Quantification of CD206−-CD11c- MΘ cells in BAT determined by FACS analysis over the time span of 0–48 h after Eto/CL treatment. (H) Quantification of CD206−-CD11c+ MΘ cells in BAT determined by FACS analysis over the time span of 0–48 h after Eto/CL treatment. (I) Quantification of NK cells in BAT determined by FACS analysis over the time span of 0–48 h after Eto/CL treatment. (J) Quantification of CD4+ cells in BAT determined by FACS analysis over the time span of 0–48 h after Eto/CL treatment. (K) Quantification of CD8+ cells in BAT determined by FACS analysis over the time span of 0–48 h after Eto/CL treatment. (L) Quantification of B cells in BAT determined by FACS analysis over the time span of 0–48 h after Eto/CL treatment. (M) Volcano plot of BAT RNA sequencing data comparing Eto/CL- versus mock/mock-treated mice (n = 4). RNAseq data were analyzed by Novogene, which is based on DESeq2 for dataset analysis. A full list of differentially expressed genes with adjusted P value < 0.01 is provided in Dataset EV1. (N) Gene ontology enrichment analysis of RNA sequencing data. The 15 most significantly altered pathways are shown. A full list of significantly altered pathways is provided in Dataset EV1. Data are presented as mean values ± SEM. *P < 0.05 by Student's t test; data were logarithmized for statistical analysis (A–D). N values indicate biological replicates. Source data are available online for this figure.

expression changes in purinergic receptors activated by ATP during BAT degeneration and inflammation. The expression of both *P2rx4* and *P2rx7* but not other ATP-activated purinergic receptors was higher in BAT of Eto/CL-treated mice compared to the other treatment groups (Fig. 3J). In particular, *P2rx5* was nearly absent in degenerated BAT (Fig. 3J), which is in line with its role in brown adipocyte differentiation and expression of thermogenic genes (Jaeckstein et al, 2025; Razzoli et al, 2024). A similar pattern for *P2rx4* and *P2rx7* was observed in BAT of sham-operated but not denervated *Ucp1*−/− mice (Fig. 3K), further strengthening the association of imbalanced thermogenic activation with purinergic receptor signaling. To determine the cell-type specific expression of these receptors, BAT samples were sorted using immune-magnetic beads. Compared to mature adipocytes and CD31-positive endothelial cells, highest expression of both *P2rx4* and *P2rx7* was detected in CD11b-positive myeloid cells (Fig. 3L). In line, re-analysis of a snRNAseq dataset (Behrens et al, 2025) indicated that only the myeloid cell fraction displayed a substantial percentage of *P2rx4/P2rx7* double positive nuclei isolated from murine BAT (Fig. EV3B), which further substantiate the role of myeloid cells in BAT degeneration. Importantly, the ATP-dependent activation of P2X4 and P2X7 results in potassium efflux and subsequent stimulation of the NLRP3 inflammasome, which is associated with higher caspase 1 activity (de Rivero Vaccari et al, 2014; Vande Walle and Lamkanfi, 2024). It is of note that the number of myeloid cells displaying high caspase activity increased in BAT of mice in response to Eto/CL treatment (Fig. 3M), suggesting an involvement of purinergic receptors and inflammasome activation in BAT degeneration.

## Genetic variants in P2RX4 and P2RX7 correlate with metabolic phenotypes

To assess the potential relevance in humans, the expression of these receptors was determined in human BAT samples that were cell-sorted by immune-magnetic beads into fractions containing mature adipocytes, CD31-positive endothelial cells and CD11b-positive myeloid cells in a similar way as described above for the murine BAT samples. The efficient sorting of the cell fractions was confirmed by enrichment of the myeloid cell marker *EMR1*

(Fig. EV3C), of the capillary endothelial cell marker *GPIHBP1* (Fig. EV3D), and of adiponectin, an adipocyte-specific hormone encoded by *ADIPOQ* (Fig. EV3E). Expression of *UCP1* in the mature adipocyte fraction indicated the presence of thermogenic adipocytes (Fig. EV3F). Similar to murine BAT, the majority of *P2RX4* and *P2RX7* was detected in the CD11b-positive fraction and myeloid cells of human BAT, respectively (Figs. 3N and EV3G).

The potential relevance of P2X receptor family members in metabolic diseases was analyzed by using the ExPheWAS browser, which is based on phenotypic and genetic data from the UK Biobank. This tool aggregates common variants within gene regions to increase statistical power compared to single SNP analysis (Legault et al, 2022). Notably, genetic variants in the coding regions of both *P2RX4* and *P2RX7* (individual SNPs are presented in Dataset EV2), but no other members of the P2X family, correlate with metabolic phenotypes, including inflammatory markers such as C-reactive protein (Fig. 3O). Furthermore, obesity-associated phenotypes such as body fat percentage, body mass index, waist circumference and even basal metabolic rate suggest an unexpected role for P2X4 and P2X7 in chronic inflammatory metabolic diseases in humans.

## Mice lacking both P2X4 and P2X7 activity are protected from BAT degeneration

Given the potential involvement of purinergic signaling in inflammatory BAT degeneration, we first studied mice lacking either *P2rx4* or *P2rx7* in the context of imbalanced thermogenic activation. Compared to wild-type controls, inflammatory responses in BAT were unaltered in *P2rx4*−/− as well as *P2rx7*−/− mice as determined by gene expression and histological analysis (Fig. EV4A–C), indicating that the two receptors might compensate for each other. To generate a model that lacks function of P2X4 and P2X7, a half-life extended camelid-derived antibody (nanobody) blocking P2X7 activity (Danquah et al, 2016) was administered to *P2rx4*−/− and littermate control mice. Notably, blockade of both receptors blunted macrophage abundance and fibrosis in BAT in response to Eto/CL treatment (Figs. 4A and EV4D–I). In line, gene expression of inflammatory markers and MAC2 protein levels were decreased, whereas UCP1 mRNA and protein levels were

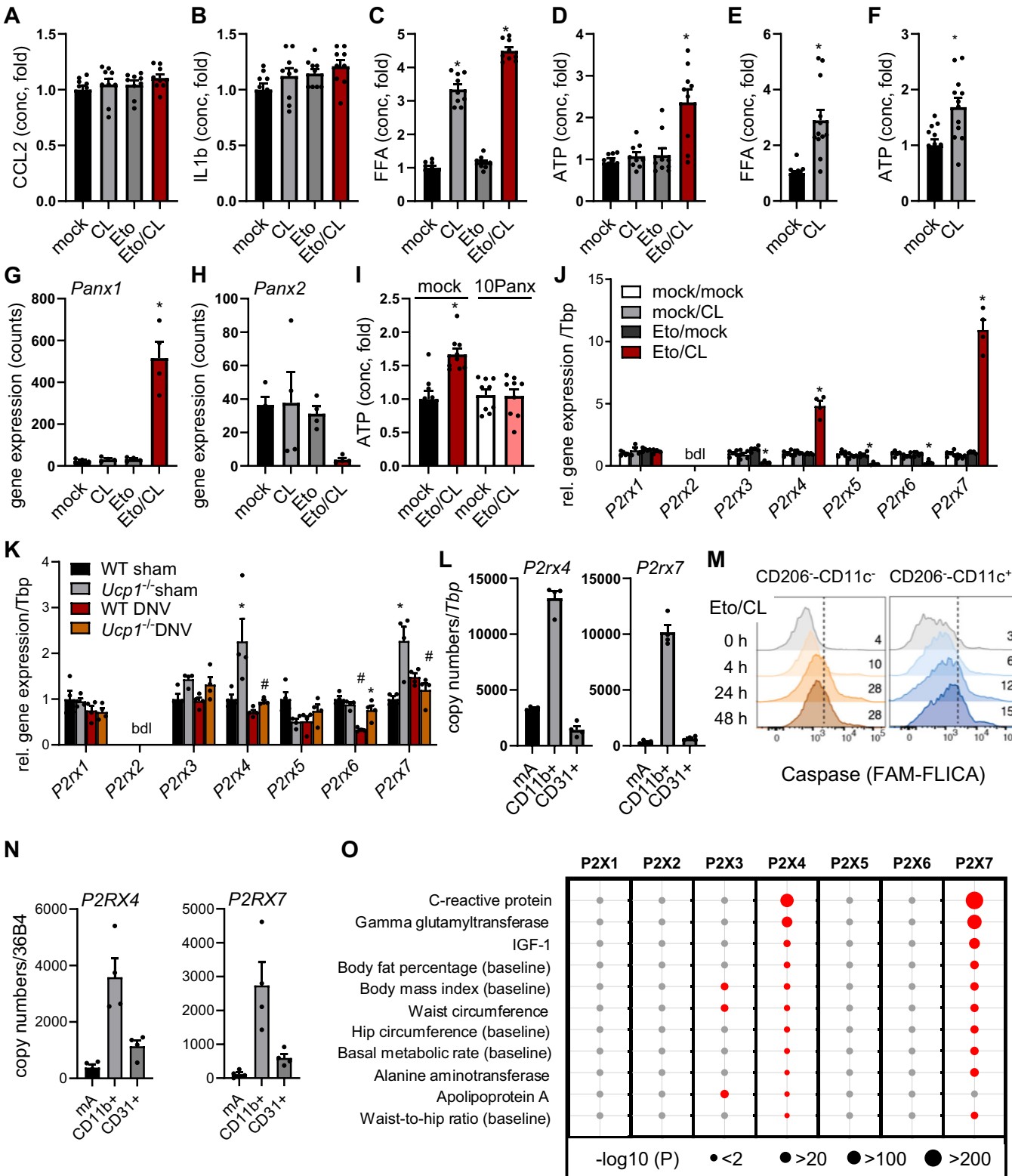

**Figure 3.  Brown adipocyte dysfunction causes ATP secretion and is linked to purinergic receptor signaling in BAT.**

(A–D) Brown adipocytes differentiated from BAT stromal vascular fractions of WT mice were treated for 24 h with or without Eto (25 μM) in the presence or absence of CL (50 nM). (A) CCL2 concentration in cell culture supernatant ($n = 9$). (B) IL1β concentration in cell culture supernatant ($n = 9$). (C) Free fatty acid (FFA) concentration in cell culture supernatant ($n = 9$). (*$P = <0.0001$, *$P = <0.0001$; left to right). (D) ATP concentration in cell culture supernatant ($n = 9$). (*$P = <0.0001$). (E, F) Brown adipocytes differentiated from BAT stromal vascular fractions of $Ucp1^{-/-}$ mice were treated for 24 h with or without CL (50 nM). (E) FFA concentration in cell culture supernatant ($n = 12$). (*$P = 7.38E-05$). (F) ATP concentration in cell culture supernatant ($n = 12$). (*$P = 0.0023$). (G, H, J), Wild-type mice housed at 22 °C were injected daily for three subsequent days with either vehicle (mock/mock), with the β3-adrenergic agonist CL316,243 alone (mock/CL), with the inhibitor of β-oxidation etomoxir alone (Eto/mock) or with the combination of both (Eto/CL). (G) Gene expression of *Panx1* in BAT ($n = 4$). (*$P = <0.0001$). (H) Gene expression of *Panx2* in BAT ($n = 4$). (I) Brown adipocytes differentiated from BAT stromal vascular fractions of WT mice were treated for 24 h with or without Eto (25 μM) and CL (50 nM) in the presence or absence of the pannexin-1 inhibitor 10Panx (100 μM). ATP concentration in cell culture supernatant ($n = 9$). (*$P = 0.0001$). (J) Gene expression of purinergic receptors *P2rx1-P2rx7* in BAT ($n = 4–5$). (*P2rx3*: *$P = 0.0014$; *P2rx4*: *$P = <0.0001$; *P2rx5*: *$P = <0.0001$; *P2rx6*: *$P = <0.0001$; *P2rx7*: *$P = <0.0001$; left to right). (K) Gene expression of purinergic receptors in BAT of denervated (DNV) or sham-operated (sham) wild-type (WT) and $Ucp1^{-/-}$ mice ($n = 4$). (*P2rx4*: *$P = 0.0197$, #$P = 0.0151$; *P2rx6*: #$P = <0.0001$, *$P = 0.0029$; *P2rx7*: *$P = 0.0015$, #$P = 0.0058$; left to right). (L) BAT of wild-type mice was sorted using low speed centrifugation to enrich mature adipocytes (mA), followed by MACS° to isolate CD31+ endothelial cells and CD11b+ myeloid cells. Gene expression of *P2rx4* and *P2rx7* ($n = 4$). (M) Wild-type mice were treated with Eto and CL for indicated time spans. Percentage of caspase-positive CD206⁻-CD11c⁻ and CD206⁻-CD11c⁺ cells of BAT determined by flow cytometry. (N) Human BAT samples from four donors were sorted using low speed centrifugation to enrich mature adipocytes (mA), followed by MACS° to isolate CD31⁺ endothelial cells and CD11b⁺ myeloid cells. Gene expression of P2RX4 and P2RX7 in the three fractions ($n = 4$). (O) Metabolic phenotypes associated with the purinergic P2X-receptor family members as analyzed with the ExPheWas Browser. Data are presented as mean values ± SEM. (A–D, G–J) *$P < 0.05$ by ANOVA compared to control; (E, F) *$P < 0.05$ by Student's $t$ test; (K) *$P < 0.05$ by ANOVA comparing WT vs. $Ucp1^{-/-}$; and #$P < 0.05$ comparing sham vs DNV. See also related Fig. EV3. N values indicate biological replicates. Source data are available online for this figure.

substantially higher in $P2rx4^{-/-}$ mice treated with the antagonistic P2X7 nanobody (Fig. 4B–D). Moreover, by an advanced flow cytometry approach that distinguishes circulating (vascular) from tissue (extravascular) CD45⁺CD11b⁺ cells (Fig. 4E), we detected a significant drop in myeloid cells in the tissue whereas a larger fraction remained in the vascular lumen in P2X7 nanobody-treated $P2rx4^{-/-}$ mice (Fig. 4E,F). In a separate experiment, we observed that the increase of CD206-negative macrophage subsets present in BAT was blunted by dual inhibition (Fig. 4G), which altogether indicated that P2X4 and P2X7 promote the infiltration of macrophages during BAT degeneration.

Mechanistically, we found that combined inhibition of P2X4 and P2X7 prevented ATP-dependent caspase 1 activation in cultured bone marrow derived macrophages (Fig. 4H), and in macrophage subsets in BAT in vivo (Fig. 4I), suggesting that inflammasome activation in myeloid cells is dependent on P2X4/7 signaling and thereby determine pro-inflammatory alterations during degenerative BAT remodeling. In line with the resistance to BAT degeneration, $P2rx4^{-/-}$ mice treated with the P2X7-inhibiting nanobody displayed higher energy expenditure and body core temperature than $P2rx4^{-/-}$ controls in response to Eto/CL treatment (Fig. 4J–L). In summary, ATP-dependent purinergic receptor signaling via P2X4/P2X7 is mandatory for myeloid cell infiltration during BAT inflammation and degeneration. Moreover, the inhibition of both receptors maintains thermogenic function, preventing the degenerative process.

### Myeloid P2X4 expression determines BAT degeneration

Although the highest expression of P2X4 and P2X7 is observed in BAT macrophages (compare Figs. 3L,N and EV3B,G), other cell types present in adipose tissues, such as endothelial cells, could contribute to the proposed paracrine signaling pathway. To investigate the relevance of P2X4 specifically in myeloid cells, we generated a cell type-specific P2X4 knockout by crossing the LysM-Cre line with floxed *P2rx4* mice. The successful cell type-specific depletion of P2X4 in macrophages (Fig. 5A–C) but not in other immune cells such as mast cells (Fig. 5D,E) was detected by flow

cytometry analysis of peritoneal immune cells isolated after lavage. To generate a model lacking both P2X4 and P2X7 activity specifically in myeloid cells, control floxed littermates ($P2xr4^{fl/fl}$-LysM-Cre-) and myeloid-specific P2X4-deficient mice ($P2xr4^{fl/fl}$-LysM-Cre + ) were treated with the antagonistic P2X7 nanobody (Figs. 5F–J and EV5). Of note, the cell-type specific blockade of myeloid P2X4 and P2X7 efficiently prevented all signs of BAT degeneration such as lipid accumulation (Figs. 5F and EV5A,B), macrophage abundance (Figs. 5F and EV5C,D), fibrosis (Figs. 5F and EV5E,F), and lower UCP1 expression (Figs. 5F and EV5G) in the Eto/CL model. The absence of lipid accumulation in BAT of $P2xr4^{fl/fl}$-LysM-Cre+ cannot be explained by impaired lipolysis in white fat depots in response to adrenergic stimulation, as indicated by comparable plasma levels of glycerol and non-esterified fatty acids (Fig. EV5H,I). In line with immunohistochemistry, an approximately fivefold higher expression of UCP1 both on protein and mRNA level was detected (Fig. 5G–I). Together with the lower MAC2 protein (Fig. 5G,H) as well as inflammatory and fibrosis marker expression (Fig. 5I), this confirms a substantial role of P2X4/P2X7 expressed by myeloid cells for inflammatory BAT degeneration. The functional relevance was shown by a higher energy expenditure in response to the Eto/CL regimen (Fig. 5J), which indicates sustained adaptive thermogenic capacity in this pharmacological model of BAT degeneration in mice lacking myeloid P2X4/P2X7 activity.

### P2X4/P2X7 blockade improves systemic glucose and lipid metabolism in distinct models of BAT degeneration

The potential beneficial effect of combined inhibition of P2X4 and P2X7 on systemic metabolism was assessed in models of BAT degeneration including the here established pharmacological Eto/CL model, adaptation to thermoneutral housing and feeding an obesogenic high fat diet. In the Eto/CL model, $P2rx4^{-/-}$ mice treated with the antagonistic P2X7 nanobody displayed an improved glucose tolerance, lower insulin levels, higher uptake of radiolabeled deoxyglucose into BAT (Fig. 6A–C) and a macroscopically healthier BAT appearance (Fig. EV6A). Moreover, the

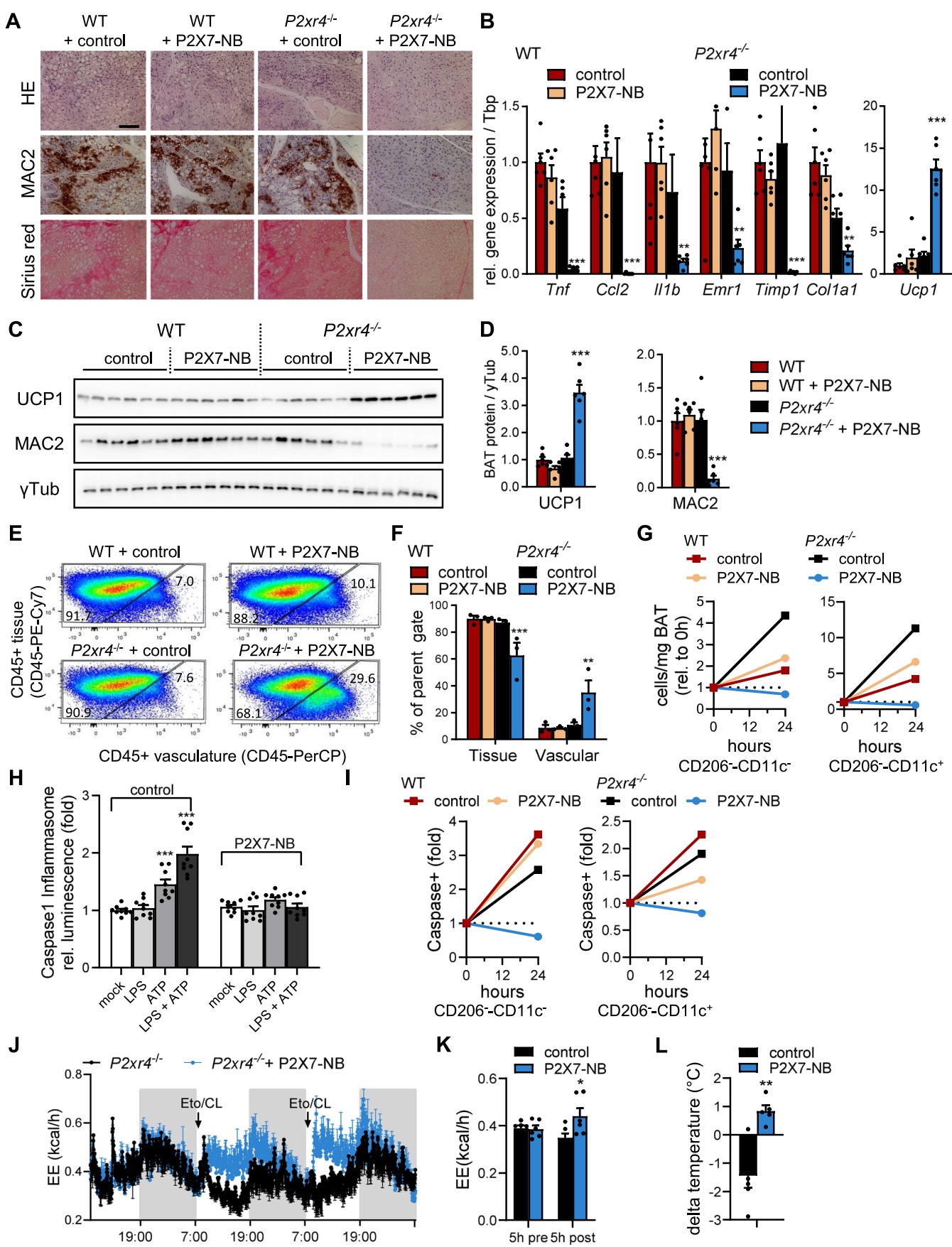

**Figure 4. Combined inhibition of P2X4 and P2X7 prevents pharmacologically induced BAT degeneration thereby preserving thermogenic capacity.**

(A–F, J–L), Wild-type (WT) and $P2rx4^{-/-}$ mice were pretreated with the P2RX7-inhibiting nanobody or vehicle (control). Then, mice were daily injected with Eto and CL on two consecutive days. (A) Representative BAT images of HE, Sirius Red and MAC2 (immune)-stainings. Scale bar, 50 μm. (B) BAT gene expression of inflammatory marker genes and $Ucp1$ ($n = 6$). ($Tnf$: *$P = 0.0006$; $Ccl2$: *$P = 0.0003$; $Il1b$: *$P = 0.0016$; $Emr1$: *$P = 0.0073$; $Timp1$: *$P = 0.0004$; $Col1a1$: *$P = 0.0053$; $Ucp1$: *$P = <0.0001$; left to right). (C) Western blot of BAT using γ-tubulin (γ-Tub) as loading control ($n = 6$). (D) Quantification of Western blot shown in (C). (UCP1: *$P = <0.0001$; MAC2: *$P = <0.0001$). (E) Representative FACS analysis of BAT-derived SVF showing vascular and infiltrated CD45$^+$-CD11b$^+$ cells. (F) Quantification of FACS analysis ($n = 3$, each pool contains BAT of two mice). (*$P = 0.0108$; *$P = 0.0112$, left to right). (G–I) Wild-type and $P2rx4^{-/-}$ mice were pretreated with the P2X7-inhibiting nanobody or vehicle (control). Then, the mice received Eto and CL for indicated time spans (0 h, 24 h). (G) Quantification of CD206$^-$-CD11c$^-$ and CD206$^-$-CD11c$^+$ cells in BAT as determined by flow cytometry. (H) Caspase-1 inflammasome activity determined in $P2rx4^{-/-}$ bone marrow-derived macrophages. Cells were pretreated with vehicle (control) or P2X7-inhibiting nanobody for 3 h. Then, macrophages were incubated without (mock), with LPS (1 μg/mL), ATP (300 μM) or a combination of LPS and ATP ($n = 9$). (*$P = 0.0004$; *$P = <0.0001$, left to right). (I) Relative increase of caspase-positive of CD206$^-$-CD11c$^-$ and CD206$^-$-CD11c$^+$ cells present in BAT determined by flow cytometry. (J) Energy expenditure (EE) of $P2rx4^{-/-}$-mice receiving either P2X7-inhibiting nanobody or vehicle. Injections of Eto/CL indicated with arrows ($n = 6$). (K) Quantification of EE for 5 h pre and post after Eto/CL injection ($n = 6$). (*$P = 0.0373$). (L) Body temperature difference 5 h pre and post after Eto/CL injection ($n = 6$). (*$P = 0.0014$). Data are presented as mean values ± SEM. (B, D, F, H), *$P < 0.05$ by ANOVA compared to WT control or mock; (K, L), *$P < 0.05$ by Student's $t$ test. N values indicate biological replicates. See also Fig. EV4. Source data are available online for this figure.

inhibition of P2X4 and P2X7 resulted in lower plasma triglycerides (Fig. 6D), which can be explained by higher disposal of $^{14}$C-triolein-labeled triglyceride-rich lipoproteins into BAT (Fig. EV6B).

Next, we investigated the effect of P2X4 and P2X7 for inflammatory processes in a more physiological setting of BAT degeneration. For this purpose, mice were housed at thermoneutrality, which closely mimics the environment of most modern-day individuals (Fischer et al, 2018; Fischer et al, 2019a) and results in a BAT phenotype similar to that observed in humans (de Jong et al, 2019). Gene expression levels of $P2rx4$ and $P2rx7$ were upregulated in BAT of mice chronically housed at thermoneutrality compared to an ambient temperature of 22 °C (Fig. EV6C), suggesting an involvement of these receptors also in BAT degeneration induced by thermoneutrality. To address a potential causal role of purinergic signaling in this process, $P2rx4^{-/-}$ mice were treated with vehicle or the P2X7-inhibiting nanobody. These mice were housed at 30 °C for 2 weeks, as this time point during adaptation to thermoneutrality is characterized by dynamic BAT remodeling including lipid accumulation and macrophage infiltration (Fischer et al, 2020b). Analysis of differentially expressed genes determined by RNAseq revealed up-regulation of thermogenic markers such as deiodinase 2 ($Dio2$), elongation of very long chain fatty acids ($Elovl3$), ATPase sarcoplasmic/endoplasmic reticulum Ca$^{2+}$ transporting 2 ($Atp2a2$), glycerol kinase ($Gk$), $Ucp1$ and phosphodiesterase 3a ($Pde3a$) and down-regulation of leptin ($Lep$) typically expressed at higher levels by lipid laden white adipocytes (Fig. 6E; Dataset EV3) (Friedman, 2019). GO enrichment analysis revealed a large number of biological processes, cellular components and molecular functions critical for tissue remodeling (Dataset EV3). Remarkably, these include several pathways that regulate ubiquitin-dependent catabolic processes and proteasomal degradation, which have been shown to be important for BAT homeostasis (Bartelt et al, 2018). These results indicate that P2X4/P2X7 signaling promotes BAT whitening in response to thermoneutral housing. In line, gene expression of thermogenic markers as well as UCP1 protein levels were higher in BAT after inhibition of both P2X4 and P2X7 (Figs. 6F,G and EV6D). Furthermore, this intervention led to reduced lipid accumulation (Figs. 6H and EV6E,F), lower triglyceride levels in BAT and plasma as determined by quantitative lipidomics (Fig. 6I–M) as well as a trend towards lower number of MAC2-positive cells (Figs. 6H and EV6G,H). Overall, these data demonstrate that purinergic signaling via P2X4 and P2X7 is

critically involved in the degeneration of BAT in response to thermoneutrality.

Obesity induced by high fat diet (HFD) feeding, in particular in combination with thermoneutral housing is also characterized by BAT degeneration (de Jong et al, 2019). For sustained inhibition of P2X7 in long-term studies, we generated an adeno-associated virus (AAV) that leads to the expression of the antagonistic P2X7 nanobody in the liver, and the subsequent secretion into systemic circulation. To allow sufficient expression, the AAV-P2X7-NB or an AAV leading to the expression of a control nanobody were injected 2 weeks prior to HFD feeding. An improved glucose tolerance was observed after 16 weeks of HFD feeding (Fig. 6N), and in the combined setup of HFD feeding and thermoneutral housing for 4 weeks (Fig. 6O). Finally, the relevance of P2X4 expressed by myeloid cells in the context of thermoneutral-induced BAT degeneration was investigated. Similar to the global knockout, we observed higher expression of typical thermogenic genes in BAT including $Ucp1$, $Dio2$ and $Ppargc1a$ in $P2xr4^{fl/fl}$-LysM-Cre+ mice that were infected with the AAV-P2X7-NB (Fig. 6P). This translated to higher UCP1 protein levels (Fig. 6Q,R) and was associated with a trend towards lower lipid accumulation and macrophage abundance (Fig. EV6I–L). In particular, improved glucose tolerance, increased uptake of radiolabeled deoxyglucose into BAT and lower plasma triglycerides (Fig. 6S–U) indicated that myeloid P2X4/P2X7 affects local and systemic metabolism by promoting BAT degeneration in response to thermoneutral housing.

## Discussion

Adaptive thermogenesis mediated by BAT is an important determinant of whole-body energy expenditure and systemic metabolism (Sakers et al, 2022). Activated thermogenic adipocytes have the ability to combust large amounts of glucose and fatty acids delivered by triglyceride-rich lipoproteins, and are thus critical for metabolic homeostasis, at least in rodents (Bartelt et al, 2011; Heine et al, 2018; Stanford et al, 2013). A number of studies detecting active BAT in healthy individuals indicates a prominent role of BAT for systemic metabolism also in humans (Chondronikola et al, 2014; Chondronikola et al, 2016; Cypess et al, 2009; Nedergaard et al, 2007; Saito et al, 2009; Scheele and Wolfrum, 2020; Virtanen et al, 2009). Supporting this view, low BAT mass and activity are

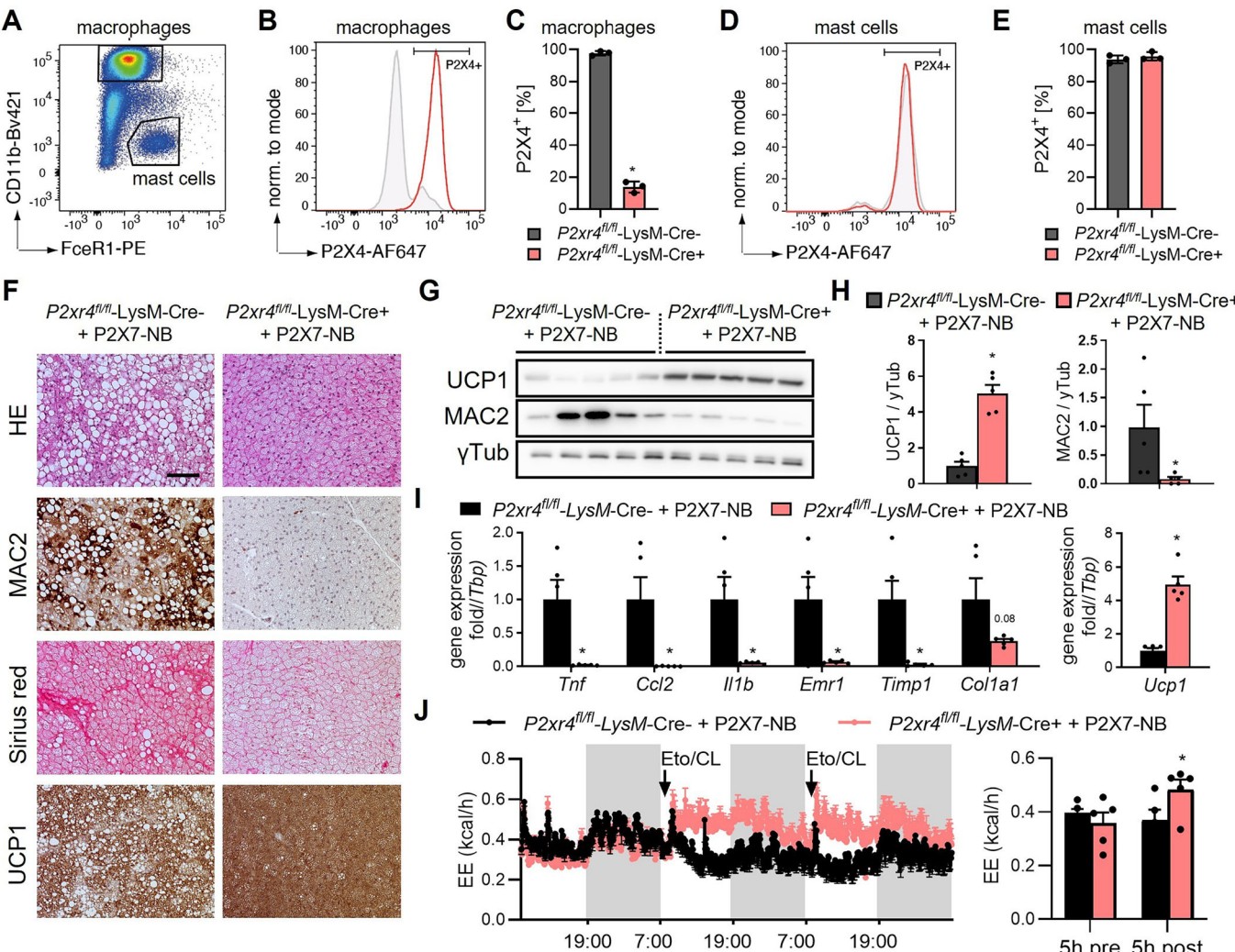

**Figure 5. Myeloid P2X4 expression determines BAT degeneration.**

(A–E) Flow cytometry-based validation of the newly generated *P2rx4*[fl/fl]-LysM[Cre] mouse model. (A) Gating of macrophages and mast cells in BAT. (B) Flow cytometry of P2X4 in peritoneal macrophages after lavage of *P2rx4*[fl/fl]-LysM[Cre-] (red line) and *P2rx4*[fl/fl]-LysM[Cre+] (grey line) mice. (C) Quantification of P2X4[+] macrophages of *P2rx4*[fl/fl]-LysM[Cre-] and *P2rx4*[fl/fl]-LysM[Cre+] mice shown in (B). (*P = <0.0001). (D) Flow cytometry of P2X4 in peritoneal mast cells after lavage of *P2rx4*[fl/fl]-LysM[Cre-] (red line) and *P2rx4*[fl/fl]-LysM[Cre+] (grey line) mice. (E) Quantification of P2X4[+] mast cells of *P2rx4*[fl/fl]-LysM[Cre-] and *P2rx4*[fl/fl]-LysM[Cre+] mice shown in (D). (F–J) *P2rx4*[fl/fl]-LysM[Cre-] and *P2rx4*[fl/fl]-LysM[Cre+] mice were pretreated with the P2X7-inhibiting nanobody. Then, mice were daily injected with Eto and CL on two consecutive days. (F) Representative BAT images of HE, Sirius Red, UCP1 and MAC2 (immune)-stainings. Scale bar, 50 μm. (G) Western blot of BAT using γ-tubulin (γ-Tub) as loading control (n = 5). (H) Quantification of Western blot shown in (G). (UCP1: *P = 6.0905E-05; MAC2: *P = 0.0469). (I) BAT gene expression of inflammatory and fibrosis marker genes and *Ucp1* (n = 5). (*Tnf*: *P = 0.0105; *Ccl2*: *P = 0.0179; *Il1b*: *P = 0.0232; *Emr1*: *P = 0.0246; *Timp1*: *P = 0.0090; *Ucp1*: *P = 4.534E-05; left to right). (J) Energy expenditure (EE) of *P2rx4*[fl/fl]-LysM[Cre-] and *P2rx4*[fl/fl]-LysM[Cre+] mice receiving P2X7-inhibiting nanobody. Injections of Eto/CL indicated with arrows and quantification of EE for 5 h pre and post Eto/CL injection (n = 5). (*P = 0.0038). Data are presented as mean values ± SEM. *P < 0.05 by Student's t test. N values indicate biological replicates. See also Fig. EV5. Source data are available online for this figure.

strongly associated with obesity, dyslipidemia, type 2 diabetes and cardiovascular diseases (Becher et al, 2021; Cypess et al, 2009; van Marken Lichtenbelt et al, 2009). Thermoneutral environments and overnutrition are typical living conditions for humans that lead to BAT degeneration characterized by profound infiltration of immune cells as well as loss of thermogenic function in mice and possibly in humans (de Jong et al, 2019; Fischer et al, 2018, 2019a; Kotzbeck et al, 2018; Villarroya et al, 2018). Not only thermo-neutral housing and energy overload, but also inefficient thermo-genesis due to loss of UCP1 is accompanied by inflammatory

changes in murine BAT (Bond et al, 2018; Kazak et al, 2017). Here, we describe a pharmacological intervention that phenocopies the pro-inflammatory tissue remodelling in UCP1-deficient mice by creating an imbalance between intracellular lipolysis, mitochondrial fatty acid respiration and thermogenesis specifically in BAT. In this model, impaired adaptive thermogenesis was associated with high expression of cytokines and immune cell markers, macrophage infiltration as well as fibrosis, pointing towards a crosstalk between brown adipocytes and immune cells in the context of BAT dysfunction. Thus, this newly established pharmacological model

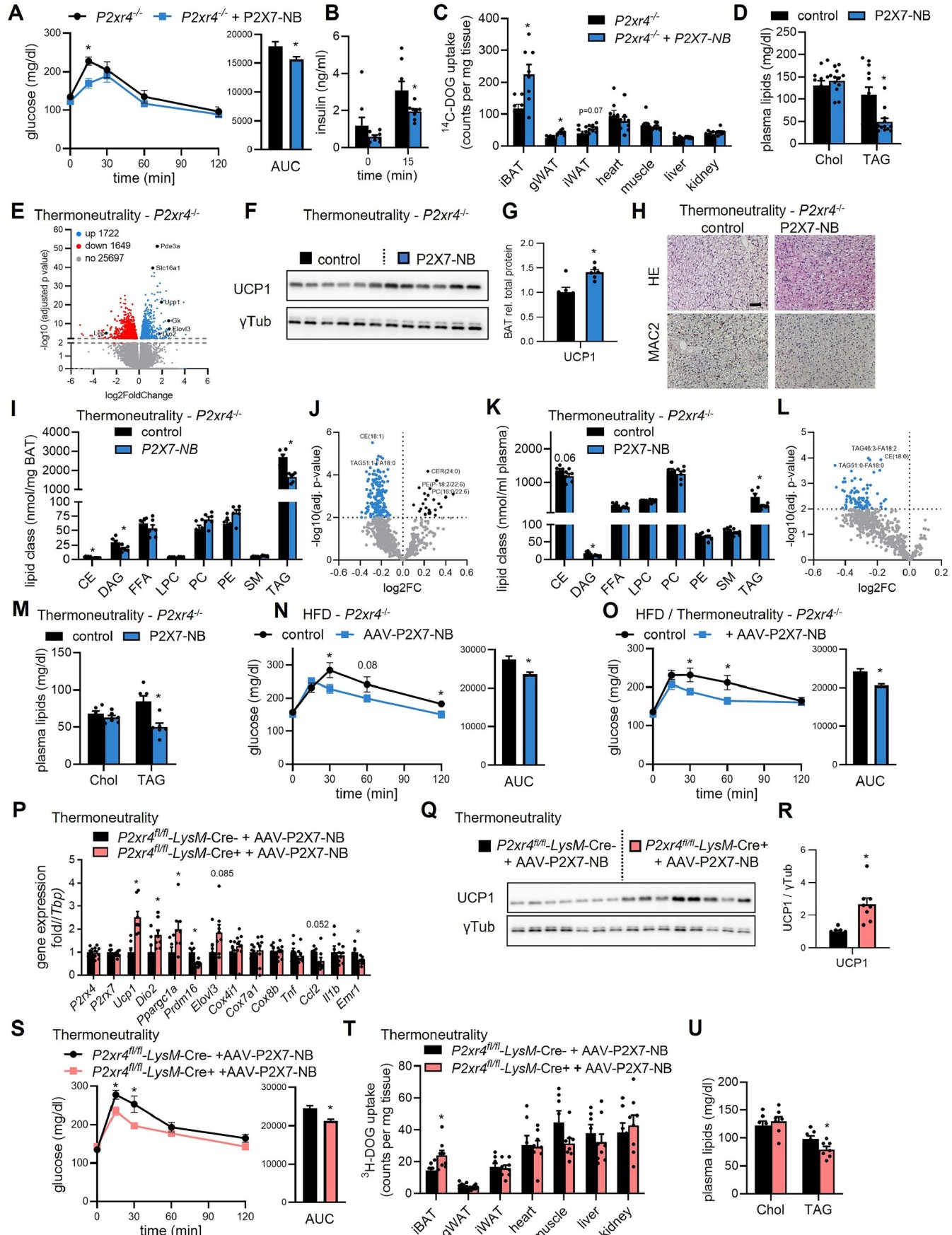

◄

**Figure 6. P2X4/P2X7 blockade improves systemic glucose and lipid metabolism in distinct models of BAT degeneration.**

(A–D) $P2rx4^{-/-}$ mice were pretreated with the P2X7-inhibiting nanobody or vehicle. Then, mice were daily injected with Eto and CL on two consecutive days. (A) Blood glucose level during oral glucose tolerance test and area under the curve (AUC) quantification ($n = 8$). (*$P = 0.0037$; *$P = 0.0226$). (B) Plasma insulin level after 0 min and 15 min of glucose challenge shown in (A) ($n = 8$). (*$P = 0.0482$). (C) Uptake of $^{14}$C-deoxyglucose (DOG) into various organs per mg tissue ($n = 8$). (*$P = 0.0062$; *$P = 0.0004$; left to right). (D) Plasma cholesterol (Chol) and triacylglyceride (TAG) levels ($n = 12$). (*$P = 0.0039$). (E–M) $P2rx4^{-/-}$ mice receiving P2X7-inhibiting nanobody or vehicle (control) were housed at thermoneutrality (30 °C) for 2 weeks. (E) Volcano plot of BAT RNA sequencing data comparing control versus P2X7-inhibiting nanobody ($n = 6$). A full list of differentially expressed genes with adjusted $P$ value < 0.01 is provided in Dataset EV3. (F) Western blot of BAT using γ-tubulin as loading control ($n = 6$). (G) Quantification of Western blot shown in (F). (*$P = 0.0058$). (H) Representative BAT images of HE, and MAC2 (immune)-stainings. Scale bar, 50 µm. (I) Lipidomic analysis of lipid classes in BAT ($n = 6$). (*$P = 2.841E-05$; *$P = 0.0186$; *$P = 0.0002$; left to right). (J) Volcano plot of lipid species in BAT comparing control versus P2X7-inhibiting nanobody ($n = 6$). (K) Lipidomic analysis of lipid classes in plasma ($n = 6$). (*$P = 0.0109$; *$P = 0.0156$; left to right). (L) Volcano plot of lipid species in plasma comparing control versus P2X7-inhibiting nanobody ($n = 6$). (M) Plasma cholesterol (Chol) and triacylglyceride (TAG) levels ($n = 6$). (*$P = 0.0032$). (N) $P2rx4^{-/-}$ mice were injected with an AAV encoding for the P2X7-inhibiting nanobody (AAV-P2X7-NB) or control vector. Two weeks after infection, the mice were fed a high fat diet (HFD) for 16 weeks. Blood glucose level during oral glucose tolerance test and area under the curve (AUC) quantification ($n = 9$–11). (*$P = 0.0413$; *$P = 0.0365$; *$P = 0.0012$; left to right). (O) $P2rx4^{-/-}$ mice were injected with an AAV encoding for the P2X7-inhibiting nanobody (AAV-P2X7-NB) or control vector. Two weeks after infection, mice were housed at thermoneutrality (30 °C) and fed a HFD for 4 weeks. Blood glucose level during oral glucose tolerance test and area under the curve (AUC) quantification ($n = 13$–14). (*$P = 0.0370$; *$P = 0.0257$; *$P = 0.0002$; left to right). (P–U) $P2rx4^{fl/fl}$-LysM$^{Cre-}$ and $P2rx4^{fl/fl}$-LysM$^{Cre+}$ mice received an AAV encoding for the P2X7-inhibiting nanobody (AAV-P2X7-NB) or control vector. Two weeks after infection, mice were housed at thermoneutrality (30 °C) for 2 weeks. (P) BAT gene expression of inflammatory and thermogenic marker genes ($n = 7$–8).). (Ucp1: *$P = 0.0005$; Dio2: *$P = 0.0498$; Ppargc1a: *$P = 0.0307$; Prdm16: *$P = 0.0006$; Emr1: *$P = 0.0027$; left to right). (Q) Western blot of BAT using γ-tubulin as loading control ($n = 7$–8). (R) Quantification of Western blot shown in (Q). (*$P = 0.0020$). (S) Blood glucose level during oral glucose tolerance test and area under the curve (AUC) quantification ($n = 7$–8). (*$P = 0.0171$; *$P = 0.0191$; *$P = 0.0008$; left to right). (T) Uptake of $^3$H-deoxyglucose (DOG) into various organs per mg tissue ($n = 7$–8). (*$P = 0.0286$). (U) Plasma cholesterol (Chol) and triacylglyceride (TAG) levels ($n = 7$–8). (*$P = 0.0378$). Data are presented as mean values ± SEM. *$P < 0.05$ by Student's $t$ test. $N$ values indicate biological replicates. See also Fig. EV6. Source data are available online for this figure.

phenocopies several important aspects of UCP1 deficiency, thermoneutral housing and high fat diet feeding, all established pathophysiological models to study BAT degeneration. In all these models, including the pharmacological one, the imbalance between energy combustion and fatty acid handling induce mitochondrial and cellular stress responses that leads to a degenerative process associated with impaired thermogenesis, lipid accumulation, immune cell infiltration and fibrosis. Despite the limitation of employing a pharmacological regimen, the advantage of the new Eto/CL model is the faster and stronger response compared to thermoneutral housing or high fat diet feeding. This enabled us to uncover the kinetics and the underlying molecular mechanisms that were confirmed in the pathophysiological models triggered by thermoneutral housing, high fat diet feeding and the combination of both, which better reflect BAT degenerative processes in humans.

Using GO enrichment analysis, we found that in addition to inflammatory pathways, ATP and purine nucleotide metabolism are significantly altered during BAT degeneration. Mechanistically, we provide strong evidence that brown adipocytes that are subjected to imbalanced thermogenic activation secrete ATP. This acts as a paracrine factor that triggers inflammatory signalling via the purinergic receptors P2X4 and P2X7 expressed by neighbouring, tissue-resident macrophages. Notably, combined inhibition of these purinergic receptors prevents BAT degeneration elicited pharmacologically or by thermoneutral housing. These data emphasize the relevance of ATP-mediated paracrine crosstalk between brown adipocytes and myeloid cells for degenerative BAT remodelling.

Our observation that ATP-sensing receptors are critically involved in the studied model systems suggests a common underlying role for the release of ATP by brown adipocytes in the course of BAT degeneration. By BAT denervation, we demonstrate that sympathetic activation is essential for the pro-inflammatory BAT remodelling observed in cold-acclimated $Ucp1^{-/-}$ mice. In a similar manner, β-adrenergic receptor signalling in

adipocytes is mandatory for BAT inflammation in mice with suppressed mitochondrial fatty acid oxidation. In both cases, fatty acids are liberated from lipid droplets by lipolysis but cannot be efficiently combusted, which is likely to induce intracellular lipotoxicity associated with ER stress and mitochondrial damage (Bond et al, 2018; Kazak et al, 2017). Recently, it has been shown that adipocytes secrete ATP in an active manner via pannexin-1 channels in a process instigated by β-adrenergic receptor signalling (Senthivinayagam et al, 2021; Tozzi et al, 2020). In our experiments, β3-adrenergic stimulation alone was not sufficient to evoke ATP secretion but required simultaneous inhibition of mitochondrial β-oxidation, arguing that intracellular accumulation of non-esterified fatty acids is critical for ATP release. In line with this notion, it has been shown that fatty acid overload causes ATP release from renal cells via mitochondrial stress-activated pannexin-1 (Sun et al, 2020). Consistent with these data, we show that the active release of ATP by stressed adipocytes is blocked by pannexin-1 inhibition. Altogether, these data imply that ATP release by brown adipocytes is primarily caused by inefficient mitochondrial lipid oxidation and could represent a common response of brown adipocytes that initiates degenerative tissue remodelling. Of note, as a unifying mechanism this would also apply to BAT degeneration under conditions of thermoneutrality and overnutrition, both characterized by inefficient lipid combustion, mitochondrial dysfunction, lipid accumulation and pro-inflammatory remodelling (de Jong et al, 2019; Schlein et al, 2021).

Among the ATP-responsive purinergic receptors (Linden et al, 2019), we found that only P2X4 and P2X7 were significantly upregulated in response to BAT degeneration, and both were predominantly expressed in macrophage fractions isolated from murine and human BAT, consistent with a recent study investigating mice (Tian et al, 2020). Of note, only combined inhibition of these purinergic receptors but not genetic deficiency of one receptor protected from BAT inflammation and loss of thermogenic function. Together with their overlapping expression pattern, these data suggest mutual compensation of P2X4 and P2X7

in the process of BAT degeneration. Further support for a compensatory mechanism of these purinergic receptors arises from previous studies demonstrating that P2X7 is dispensable for energy and metabolic homeostasis of adipose tissues in response to high fat diet feeding or cold exposure (Sun et al, 2012; Tian et al, 2020). Moreover, we were able to block ATP-dependent caspase 1 activation in P2X4-deficient macrophages both in vitro and in vivo, which were pretreated with a P2X7-inhibitory nanobody. This is consistent with the established role of P2X4 and P2X7 in NLRP3 inflammasome activation and cytokine production described in vitro (Hung et al, 2013; Karmakar et al, 2016; Kawano et al, 2012; Sakaki et al, 2013). In this context it is of note that inflammasome activation in white adipose tissue of obese mice, a site of pronounced inflammatory and fibrotic remodelling associated with insulin resistance and other metabolic disturbances (Crewe et al, 2017), is not affected by P2X7-deficiency (Sun et al, 2012).

As both P2X4 and P2X7 are highly upregulated in murine adipose tissues in response to high fat diet feeding (Tian et al, 2020) and thermoneutrality as shown in the current study, we demonstrated that combined P2X4 and P2X7 inhibition on myeloid cells protects against obesity-associated metabolic derangements such as insulin resistance, impaired glucose tolerance and dyslipidemia. Moreover, the beneficial effects of P2X4/P2X7 blockade on systemic metabolism are mediated by maintaining metabolic function of BAT. Although the human relevance and in particular the tissue-specific effects of these findings can only be addressed in intervention studies, it is remarkable that genetic variants of both *P2RX4* and *P2RX7* correlate with metabolic phenotypes including markers for metabolic inflammation, body fat and basal metabolic rate. By regulating adipose plasticity in response to local inflammation, our data suggest that myeloid P2X4/P2X7 are new players determining phenotypic, metabolic and structural alterations in response to environmental challenges known to trigger adipose tissue dysfunction (Sakers et al, 2022).

ATP can be either co-released with norepinephrine by sympathetic nerves innervating BAT that express the vesicular nucleotide transporter VNUT (Razzoli et al, 2016), or by apoptotic brown adipocytes observed under conditions of thermoneutrality (Niemann et al, 2022). In the extracellular environment, ATP is rapidly converted to adenosine and inosine, which have been shown to activate thermogenic responses in WAT and BAT (Gnad et al, 2022; Gnad et al, 2014; Niemann et al, 2022). Our data indicate that extracellular ATP released by brown adipocytes is sensed by tissue-resident macrophages that propagate the signal by releasing pro-inflammatory cytokines and chemokines, leading to the recruitment of myeloid cells and T cells. A possible explanation is that under conditions of sustained stress, the high concentration of extracellular ATP might exceed the conversion capacity, and thus ATP can directly serve as paracrine factor. Next to ATP that is often described as danger signal released in response to cell stress or death (Linden et al, 2019), an additional signal is required for the formation and activation of the NLRP3 inflammasome (Prochnicki and Latz, 2017). This leads to the question of which factor could be responsible for macrophage inflammasome activation in degenerating BAT. A unique feature of adipocytes is the ample lipolytic release of non-esterified fatty acids (Grabner et al, 2021), which similar to lipopolysaccharides can act as pro-inflammatory signalling molecules to promote NLRP3 inflammasome (Prochnicki

and Latz, 2017). Thus, in concert with the ATP-purinergic axis, fatty acids are the likely endogenous trigger during BAT degeneration that mediate full NLRP3 inflammasome activation in tissue-resident macrophages.

We have to acknowledge that the study has several limitations. First, the pharmacological model to induce BAT degeneration is exaggerated compared to the response to thermoneutral housing. However, in other contexts pharmacological models have provided a lot of mechanistic insight in the past. For instance, methionine choline deficient diets or $CCl_4$ treatment have been widely and successfully used to study degenerative liver diseases including fibrosis. Second, we did not perform single cell RNA sequencing analysis, which would allow unbiased mapping of intercellular communications during the course of BAT degeneration. Third, we did not study double knockout mice with a genetic deletion of both *P2rx4* and *P2rx7*. Fourth, we showed that pannexin 1 mediates ATP release by stressed adipocytes in vitro. However, we did not provide in vivo evidence and therefore cannot exclude other mechanisms of ATP release, e.g. VNUT-dependent ATP release by adipocytes or other cell types.

Taken together, this study shows that pharmacological, genetic and environmental suppression of thermogenesis drives BAT inflammation in an ATP-signaling-dependent manner. Inhibition of the ionotropic purinergic receptors P2X4 and P2X7 prevents the recruitment of immune cells, subsequent inflammatory processes, fibrosis and loss of thermogenic activity. This study emphasizes the relevance of inflammatory processes for adipose tissue homeostasis in the context of metabolic disturbances and highlights the critical role of paracrine crosstalk between adipocytes and tissue-resident immune cells.

# Methods

**Reagents and tools table**

| Reagent/resource | Reference or source | Identifier or catalog number |
|---|---|---|
| **Experimental models** | | |
| Mouse: C57BL/6J | Charles River | 632C57BL/6J |
| Mouse: Balb/c.P2rx4^tm1Rass | Nolte lab (UKE) | MGI:3665297 |
| Mouse: Balb/c.P2rx7^tm1Gab | Nolte lab (UKE) | MGI:2386080 |
| Mouse: B6.Ucp1^tm1Kz | Jackson Laboratories | Stock number 003124 |
| Mouse: P2rx4^tm1c(EUCOMM)Wtsi/H | MRC Harwell | Stock ID: EM:09045 |
| Mouse: P2rx4^fl/fl-LysM^Cre | This paper | N/A |
| **Antibodies** | | |
| Rabbit polyclonal anti-AKT | Cell Signaling | Cat#9272; RRID: AB_329827 |
| Rat monoclonal anti-CD11b APC (clone M1/70) | BioLegend | Cat# 101212, RRID: AB_312795 |
| Rat monoclonal anti-CD16/CD32 | BioXCell | Cat# BE0307, RRID: AB_2736987 |
| Rat monoclonal anti-CD45 PE/Cy7 (clone 30-F11) | BioLegend | Cat# 103114, RRID: AB_312979 |
| Rat monoclonal anti-CD45 PerCP (clone 30-F11) | BioLegend | Cat# 103130, RRID: AB_893339 |
| Rat monoclonal anti-CD11b-BUV737 (clone M1/70), | BD Bioscience | Cat# 612800, RRID: AB_2738811 |
| Rat monoclonal anti-CD206-APC (clone C068C2) | Biolegend | Cat# 141707, RRID: AB_10896057 |

| Reagent/resource | Reference or source | Identifier or catalog number |
|---|---|---|
| Rat monoclonal anti-Ly6G-AF700 (clone 1A8) | Biolegend | Cat# 127621, RRID: AB_10643269 |
| Rat monoclonal anti-CD8a-BUV395 (clone 53-6.7) | BD Bioscience | Cat# 565968, RRID: AB_2732919 |
| Rat monoclonal anti-F4/80-BUV563 (clone T45-2342) | BD Bioscience | Cat# 749284, RRID: AB_2873659 |
| Rat monoclonal anti-CD4-Bv421 (clone GK1.5) | Biolegend | Cat# 100437, RRID: AB_10900241 |
| Hamster monoclonal anti-CD3e-Bv650 (clone 500A2) | BD Bioscience | Cat# 740461, RRID: AB_2740187 |
| Mouse monoclonal anti-NK1.1-Bv711 (clone PK136) | Biolegend | Cat# 108745, RRID: AB_2563286 |
| Rat monoclonal anti-B220-PE (clone RA3-6B2) | Biolegend | Cat# 103207, RRID: AB_312992 |
| Hamster monoclonal anti-CD11c-PE-Dazzle (clone N418) | Biolegend | Cat# 117347, RRID: AB_2563654 |
| Rabbit monoclonal anti-γ-Tubulin | Abcam | Cat# ab179503, RRID: N/A |
| Rat monoclonal anti-IgG1 Bv421 (clone RMG1-1) | BioLegend | Cat# 406615, RRID: AB_2562233 |
| Rabbit polyclonal anti-pHSL | Cell Signaling | Cat# 4139, RRID:AB_2135495 |
| Rabbit polyclonal anti-HSL | Cell Signaling | Cat# 4107, RRID:AB_2296900 |
| Rat monoclonal anti-MAC2 | Santa Cruz | Cat# sc-23938, RRID: AB_627658 |
| Rabbit monoclonal anti-PLIN1 (D1D8) | Cell Signaling | Cat# 9349, RRID:AB_10829911 |
| Rabbit recombinant anti-TH | R&D | Cat# ab137869, RRID: AB_2801410 |
| Rabbit polyclonal anti-UCP1 | Abcam | Cat# ab10983, RRID:AB_2241462 |
| Mouse monoclonal anti-UCP1 | R&D | Cat# MAB6158, RRID: AB_10572490 |
| Anti-ARTC2.2 (clone s + 16-dimer) | Nolte lab (UKE) | Rissiek et al, 2014 |
| Anti-HLE mIgG1 (clone mab77) | Nolte lab (UKE) | N/A |
| Anti-P2X7-HLE NB (clone 13A7-dimer half life extended) | Nolte lab (UKE) | Danquah et al, 2016 |
| HRP goat anti-mouse | Jackson ImmunoResearch Labs | Cat# 115-035-003, RRID:AB_10015289 |
| HRP goat anti-rabbit | Jackson ImmunoResearch Labs | Cat# 111-035-144, RRID:AB_2307391 |
| HRP goat anti-rat | Jackson ImmunoResearch Labs | Cat# 112-035-175, RRID:AB_2338140 |
| **Oligonucleotides and other sequence-based reagents** | | |
| TaqMan assay for *ADIPOQ* | Thermo Fisher | Hs00605917_m1 |
| TaqMan assay for *Ccl2* | Thermo Fisher | Mm00441242_m1 |
| TaqMan assay for *Cd4* | Thermo Fisher | Mm00442754_m1 |
| TaqMan assay for *Cd8b1* | Thermo Fisher | Mm00438116_m1 |
| TaqMan assay for *Col1a1* | Thermo Fisher | Mm00801666_g1 |
| TaqMan assay for *Cox4i1* | Thermo Fisher | Mm00438289_g1 |
| TaqMan assay for *Cox7a1* | Thermo Fisher | Mm00438297_g1 |
| TaqMan assay for *Cox8b* | Thermo Fisher | Mm00432648_m1 |
| TaqMan assay for *Dio2* | Thermo Fisher | Mm0515664_m1 |
| TaqMan assay for *Elovl3* | Thermo Fisher | Mm00468164_m1 |

| Reagent/resource | Reference or source | Identifier or catalog number |
|---|---|---|
| TaqMan assay for *Emr1* | Thermo Fisher | Mm00802530_m1 |
| TaqMan assay for *EMR1* | Thermo Fisher | Hs00173562_m1 |
| TaqMan assay for *GPIHBP1* | Thermo Fisher | Hs01564843_m1 |
| TaqMan assay for *Il1b* | Thermo Fisher | Mm00434228_m1 |
| TaqMan assay for *Ifng* | Thermo Fisher | Mm00801778_m1 |
| TaqMan assay for *Ppargc1a* | Thermo Fisher | Mm00447183_m1 |
| TaqMan assay for *Prdm16* | Thermo Fisher | Mm00712556_m1 |
| TaqMan assay for *P2rx1* | Thermo Fisher | Mm00435460_m1 |
| TaqMan assay for *P2rx2* | Thermo Fisher | Mm00462952_m1 |
| TaqMan assay for *P2rx3* | Thermo Fisher | Mm00523699_m1 |
| TaqMan assay for *P2rx4* | Thermo Fisher | Mm00501787_m1 |
| TaqMan assay for *P2rx5* | Thermo Fisher | Mm00473677_m1 |
| TaqMan assay for *P2rx6* | Thermo Fisher | Mm00440591_m1 |
| TaqMan assay for *P2rX7* | Thermo Fisher | Mm01199500_m1 |
| TaqMan assay for *Tbp* | Thermo Fisher | Mm00446973_m1 |
| TaqMan assay for *Timp1* | Thermo Fisher | Mm00441818_m1 |
| TaqMan assay for *Tnf* | Thermo Fisher | Mm00443258_m1 |
| TaqMan assay for *Ucp1* | Thermo Fisher | Mm00494069_m1 |
| TaqMan assay for *UCP1* | Thermo Fisher | Hs00222453_m1 |
| TaqMan assay for *36B4* | Thermo Fisher | Hs99999902_m1 |
| SYBR-Primer for hP2rX4 forward | 5′-CCTCTGCTTG CCCAGGTACTC-3′ | (Lee et al, 2006) |
| SYBR-Primer for hP2rX4 reverse | 5′- CCAGGAGATA CGTTGTGCTCAA-3′ | (Lee et al, 2006) |
| SYBR-Primer for hP2rX7 forward | 5′- TCTTCGTGA TGACAAACTTT CTCAA-3′ | (Lee et al, 2006) |
| SYBR-Primer for hP2rX7 reverse | 5′-GTCCTGCG GGTGGGATACT-3′ | (Lee et al, 2006) |
| **Chemicals, enzymes and other reagents** | | |
| APC-Cy7 LIVE/DEAD™ | Thermofisher | Cat# L10119 |
| ATP | Sigma | Cat# A2383 |
| Bovine serum albumin | Sigma | Cat# A3059 |
| CD11b MicroBeads | Miltenyi | Cat# 130-049-601 |
| CD31 MicroBeads | Miltenyi | Cat# 130-097-418 |
| CD31 MicroBead kit | Miltenyi | Cat# 130-091-935 |
| CL316,243 | Tocris | Cat# 1499 |
| Collagenase II | Sigma | Cat# C2-22-BC |
| Collagenase D | Roche | Cat# 11088882001 |
| cOmplete™ Mini Protease Inhibitor Cocktail | Roche | Cat# 1836153001 |
| Dispase II | Gibco | Cat# 17105-041 |
| DMEM, high glucose | Gibco | Cat# 11965118 |
| DMEM-F12 (1:1), Glutamax | Gibco | Cat# 31331093 |
| Eosin | Merck | Cat# 1.15935 |
| Etomoxir | Sigma | Cat# E1905 |
| EUKITT | O. Kindler | O. Kindler N/A |
| Fetal Bovine Serum | Gibco | Cat# 10270-106 |
| Hematoxylin | Sigma | Cat# MHS32-1L |
| Insulin | Sigma | Cat# I9278 |
| Lipopolysaccharides from Escherichia coli O55:B5 | Sigma | Cat# L6529 |
| Murine M-CSF | Peprotech | Cat# AF-315-02-10U |

| Reagent/resource | Reference or source | Identifier or catalog number |
|---|---|---|
| Newborn calf serum (NCS) | Sigma | Cat# N4637 |
| NuPAGE LDS 4x sample buffer | Invitrogen | Cat# NP0008 |
| NuPAGE reducing sample buffer | Invitrogen | Cat# NP0004 |
| Rosiglitazone | Cayman Chemicals | Cat# 18003649897 |
| SolvableTM | PerkinElmer | Cat# 6NE9100 |
| Scintillation fluid | Zinsser Analytic | Cat# 1008500 |
| SuperSignalWest Femto Substrate | Thermo Fisher | Cat# 34095 |
| Trizol | peqlab | Cat# 30-2010 |
| $^{14}$C-deoxyglucose | PerkinElmer | Cat# NEC720A |
| $^{3}$H-deoxyglucose | Hartmann Analytik | Cat# ART0103 |
| $^{14}$C-triolein | Hartmann Analytik | Cat# ARC0291 |
| **Software** | | |
| BD FlowJo 10.8.1 | Becton Dickinson | https://www.flowjo.com/solutions/flowjo RRID:SCR_008520 |
| Excel 2016 (Version 16.16.20) | Microsoft | https://www.microsoft.com RRID:SCR_016137 |
| ExPheWas Browser (Version Browser v1.3.0 // Data v1.0) | | https://exphewas.statgen.org/ (Legault et al, 2022) |
| Graphpad Prism (Version 9.2) | Graphpad | https://www.graphpad.com RRID:SCR_002798 |
| Image Studio Lite (Version 5.2.5) | LICOR | https://www.licor.com/bio/image-studio-lite/ RRID:SCR_013715 |
| ImageJ/Fiji | NIH | https://imagej.net/ij/ RRID:SCR_003070 |
| Macro Interpreter v2.41 | Sable Systems International | https://www.sablesys.com/ |
| NIS-Elements Advances Research | Nikon | https://www.nis-elements.cz RRID:SCR_014329 |
| Promethion Live v21.0.0 | Sable Systems International | https://www.sablesys.com/ |
| LipidomicsWorkflowManager | SCIEX | RRID:SCR_017003 |
| **Other** | | |
| AA45/32 Phys Control Plasma | SCIEX | Cat# 4386703 |
| Accu-CHEK Aviva | Roche | Cat# 06453970 |
| ATP Bioluminescent Assay Kit for ATP quantitation | Sigma | Cat# FL-AA |
| Caspase-Glo® 1 Inflammasome Assay | Promega | Cat# G9951 |
| Cholesterol FS | DiaSys | Cat# 113009910704 |
| FAM-FLICA(R) Caspase 1 Assay Kit | Biomol | Cat# ICT-98 |
| Free Glycerol Reagent | Sigma | Cat# F6428 |
| G2 E-Mitter | Starr Life Sciences | |
| High Capacity cDNA RT kit | Invitrogen | Cat# 4368813 |
| High fat diet 35.5% Lard | EF Bio-Serv | Cat# F3282 |
| Internal Standards Kit for Lipidyzer platform | SCIEX | Cat# 5040156 |
| MACS® LS Column | Miltenyi | Cat# 130-042-401 |
| Mouse CCL2/JE/MCP-1 DuoSet ELISA | R&D | Cat# DY479 |

| Reagent/resource | Reference or source | Identifier or catalog number |
|---|---|---|
| Mouse IL-1 beta/IL-1F2 DuoSet ELISA | R&D | Cat# DY401 |
| NEFA-HR (2) assay | Wako/Fujifilm | Cat# 270-77000 |
| Nitrocellulose blotting membrane | GE Healthcare Amersham™Protan™ | Cat# 10600002 |
| Noradrenaline/Norepinephrine ELISA Kit | Antibodies.com | Cat# A74229 |
| NucleoSpin RNA/Protein kit | Macherey & Nagel | Cat# 740933 |
| QC Spike Standards Kit for Lipidyzer platform | SCIEX | Cat# 5040408 |
| Steady DAB/Plus | Abcam | Cat# ab103723 |
| Triglycerides FS | DiaSys | Cat# 157109910026 |
| Ultra sensitive rat insulin ELISA Kit | Crystal Chem | Cat# 90060 |
| pAAV[Exp]-TBG > (l-10edimAlb) | VectorBuilder | Cat# VB220321-1428pma |
| pAAV[Exp]-TBG > (13A7dimAlb) | VectorBuilder | Cat# VB220321-1424dbh |

## Animal models

Animal experiments were approved by the Animal Welfare Officers of University Medical Center Hamburg-Eppendorf (UKE) and Behörde für Gesundheit und Verbraucherschutz Hamburg and conducted in accordance with ARRIVE guidelines for ethical regulations and policies (reference number 0082/2020). Mouse lines used in this study are listed in the Reagents and Tools Table. Mice were raised in the animal facility of UKE at room temperature (22 °C) and held at a 12-h light, 12-h dark cycle with *ad libitum* access to standard laboratory chow diet and water. C57BL/6 mice were purchased from Charles River. $Ucp1^{-/-}$ and wild-type littermates on a C57BL/6 background as well as $P2rx4^{-/-}$, $P2rx7^{-/-}$ and wild-type controls on a Balb/c background were bred in the animal facility of the UKE. To generate $P2rx4^{fl/fl}$ mice, the sperm of C57BL/6N-$P2rx4$tm1c(EUCOMM)Wtsi/H (strain ID EM:09045) was obtained from Mary Lyon Centre at MRC Harwell (Oxfordshire, UK). For myeloid-specific deletion, $P2rx4^{fl/fl}$ mice were bred with mice carrying the LysM-Cre transgene to receive $P2rx4^{fl/fl}$-LysM$^{Cre-}$ and $P2rx4^{fl/fl}$-LysM$^{Cre+}$ mice. Age matched male mice (10–18 weeks) were used and housed at 22 °C or 30 °C. UCP1-deficient mice were gradually acclimated to 6 °C over a period of 14 days as described (Fischer et al, 2020a). For thermoneutral studies, mice were acclimated to 30 °C for 2 weeks. For high fat (HFD) feeding, mice were fed a lard-based HFD (EF Bio-Serv F3282; 35.5% lard) for 16 weeks. Combined thermoneutral housing and HFD feeding was performed for 4 weeks. Futile thermogenic activation using etomoxir/CL was performed with housed at 22 °C. For all terminal procedures, mice were fasted for 4 h and anesthetized with ketamine (180 mg/kg) and xylazine (24 mg/kg). Body weights and BAT weights of all metabolic studies are given in Dataset EV4.

## Human BAT samples

For the isolation of mature adipocytes, endothelial cells and myeloid cells from human deep-neck BAT samples (two males and two females; aged between 40 and 60) were collected during thyroid surgery at the Department of General, Visceral and Thoracic Surgery,

University Medical Center Hamburg-Eppendorf. All participants signed an informed consent, the study was approved by the Ethics Committee of the Hamburg Chamber of Physicians (PV4889) and conducted in accordance to the Declaration of Helsinki.

## Denervation of brown adipose tissue

BAT denervation was performed as described previously (Fischer et al, 2019b). Briefly, mice housed at thermoneutrality from the time of weaning received 0.2 mg/kg meloxicam one day before and the consecutive days after surgery. For surgery, mice received 5 mg/kg carprofen and were anaesthetized with 4% isoflurane inhalation in $O_2$. The area above the interscapular region was shaved and sterilized using 80% ethanol. After opening the skin in a 1 cm incision, the BAT was carefully detached from the underlying muscle layer and the nerves were prepared. The nerve fibers were cut or only exposed and touched in the sham-operated group. The animals were allowed to recover for 1 week at thermoneutrality and then transferred to room temperature.

For analysis of norepinephrine levels, brown adipose tissues were homogenized in 10x w/v RIPA buffer (50 mM Tris-HCl pH 7.4; 5 mM EDTA; 150 mM NaCl; 1 mM Na-pyrophosphate; 1 mM NaF; 1 mM Na-vanadate; 1% NP-40, cOmplete Mini Protease Inhibitor Cocktail Tablets (Roche)) using a TissueLyser (QIAGEN). After centrifugation (10 min, 13,000 rpm), supernatants were collected and assayed using the Noradrenaline/Norepinephrine ELISA Kit (A74229, antibodies.com) according to the manufacturer's protocol.

## Pharmacological interventions

Mice were daily injected for a maximum of three days with etomoxir (i.v. 10 mg/kg in 10% DMSO, 90% saline) and/or with CL316,243 (s.c. 1 mg/kg in saline). Mock controls received vehicle only. For inhibition of P2X7, the long-acting, inhibitory nanobody 13A7 (Danquah et al, 2016) was dissolved in saline and mice were i.p. injected (70 μg per mouse) 4 h prior to etomoxir-treatment or thermoneutral housing. For AAV-mediated intrinsic expression of the inhibitory nanobody 13A7 (Danquah et al, 2016), mice were i.v. injected with a recombinant AAV8 encoding for the P2X7-inhibiting nanobody or a control (titer: $1 \times 10^{11}$ GC /mouse; produced by VectorBuilder). To achieve efficient expression and secretion of the nanobody, the coding region was driven by the liver-specific thyroxine binding globulin (TBG) promoter. The design and timelines of each experiment are presented in Dataset EV4.

## Glucose and lipid uptake studies

For the postprandial challenge, mice were fasted for 2 h and afterwards received a gavage of a lipid-glucose-emulsion containing glucose (2 mg/g body weight) and triglycerides (3.7 mg/g body weight) that was traced with $^{14}C$-deoxyglucose (0.15 MBq/kg body weight). For oral glucose tolerance test (OGTT), mice orally received a glucose solution (2 mg/g body weight) that for some experiments was traced with $^3H$-deoxyglucose (0.72 MBq/kg body weight). For lipid turnover uptake studies, mice were i.v. injected with a TRL solution containing $^{14}C$-triolein (7.4 kBq per mouse) as described (Fischer et al, 2021). Blood glucose concentrations were measured in blood taken from tail vein after 0 min, 15 min, 30 min,

60 min, 120 min using AccuCheck Aviva glucose sticks (Roche). For uptake experiments, tissues were collected after transcardial perfusion with phosphate buffer saline containing 10 U/ml heparin and dissolved using SolvableTM. Radioactivities were quantified by liquid scintillation counting (Tricarb, PerkinElmer).

## Lipolysis assay

For assessing lipolysis in adipose tissues, mice were injected with CL316,243 (s.c. 1 mg/kg in saline). Tail blood samples were collected after 0 min, 30 min, 60 min and 120 min using EDTA-containing microvettes (Sarstedt; 16.444.100). FFA and glycerol levels were determined using the enzymatic NEFA-HR kit (Wako, Fujifilm) and Free Glycerol Reagent (F6428, Sigma).

## Plasma parameters

Plasma lipid levels (triglycerides and cholesterol) were measured by enzymatic-colorimetric assays (DiaSys; 157109910026; 113009910704) according to the instructions of the manufacturer. Plasma insulin levels were determined by ELISA as described in the company's protocol (Crystal Chem; 90060).

## Histochemistry

For histological staining, tissues were fixed in 3.7% formaldehyde in PBS and embedded in paraffin. Hematoxylin-Eosin (H&E) and Sirius Red stainings were performed using standard procedures. For immunhistochemistry stainings, rehydration and antigen retrieval was performed as described previously (Fischer et al, 2019b). Sections were blocked with 3% BSA for 1 h at RT. Primary antibody incubation was performed overnight at 4 °C in a humid incubation chamber. Following primary antibody were used: rat-anti-MAC2 (1:250, sc-23938, Santa Cruz), rabbit-anti-UCP1 (1:500, ab10983). Sections were washed with PBS and incubated for 1 h with the horseradish peroxidase (HRP) coupled-secondary antibody: HRP-donkey-anti-rat (1:500, 712-036-153, Jackson Immunoresearch), HRP-goat-anti-rabbit (1:500, 111-035-144, Jackson Immunoresearch). Staining was performed using a DAB kit (abcam103723) according to manufacturer's instructions. Afterwards, nuclei were counterstained with hematoxylin. Stainings were blued under running tap water for 15 min. After dehydration, slides were mounted using Eukitt. Images were taken using a NikonA1 Ti microscope equipped with a DS-Fi-U3 brightfield camera. Quantifications were performed with ImageJ (https://imagej.net/ij/).

## Isolation of human endothelial cells, myeloid cells and adipocytes

Endothelial cells and macrophages were isolated from murine and human BAT as described previously for mice (Fischer et al, 2021) with minor alterations for the human samples. Briefly, human BAT was minced, and digested for 30 min at 37 °C in isolation buffer (123 mM NaCl, 5 mM KCl, 1.3 mM $CaCl_2$, 5 mM glucose, 100 mM HEPES; pH:7.4) containing 600 U/mL collagenase II and 1.5% BSA. For murine BAT, the digestion buffer contained PBS supplemented with 10 mM $CaCl_2$, 2.4 units/mL dispase II, and 1.5 units/mL collagenase D. The digested tissue was filtered through a 100 μm cell strainer and centrifuged for 5 min at 600×g, 4 °C. The

supernatant was harvested as mature adipocyte fraction. Subsequent sorting was performed as described (Fischer et al, 2021). Following magnetic microbeads were used: mouse/human CD11b microbeads (Miltenyi; 130-049-601, 10 µl beads/$10^7$ cells), human CD31 microbead kit (Miltenyi; 130-091-935, 10 µl beads/$10^7$ cells and 10 µl FcR blocking reagent /$10^7$ cells), mouse CD31 microbeads (Miltenyi; 130-097-418, 10 µl beads/$10^7$ cells). Cell fractions were pelleted and re-suspended in TRIzol® reagent for RNA extraction.

## Cell culture

For culturing primary brown adipocytes, BAT from male and female wild-type C57BL/6 or $Ucp1^{-/-}$ mice aged 4–6 weeks was harvested, the stromal-vascular fraction was isolated and differentiated as described previously (Sass et al, 2021). After differentiation, the cells were pretreated with etomoxir (25 µM) or vehicle (DMSO) for 24 h. Afterwards, etomoxir (25 µM), CL316,243 (50 nM) or vehicle (DMSO) was added for another 24 h. Supernatants were snap-frozen or immediately used for analyses. Cells were washed with PBS and harvested in Trizol.

CCL2 and IL1β concentrations were determined in the adipocyte supernatants using ELISA (CCL2: DY479; IL1β: DY401; R&D) according to the manufacturer's instructions. FFA levels were measured using the enzymatic NEFA-HR kit (Wako, Fujifilm). For ATP determination, a bioluminescence-based kit was used (FL-AA, Sigma).

For culturing bone marrow derived macrophages, bones of the hindleg from male and female $P2rx4^{-/-}$ mice were prepared by removing muscle tissues. Tibia and femur were cut open and the bone marrow was harvested by centrifugation (1 s, 6000×g, 4 °C). The pellets were re-suspended and plated in uncoated petri dishes. Differentiation was performed in DMEM-F12 (1:1) Glutamax containing 10% FBS, 1% P/S and 30 ng/mL MCSF. On day 5, the cells were split into two dishes. On day 7, the cells were detached with ice cold PBS containing 2.5 mM EDTA (5 min) and seeded for the experiments. Macrophages were pretreated with the P2RX7-inhibiting nanobody or vehicle for 3 h. Afterwards, LPS (1 µg/mL) was added 4 h prior to ATP treatment (300 µM) for additional 4 h. Caspase-1 Inflammasome Activity was measured using a luminescence-based kit according to manufacturer's instructions (G9951, Promega).

## Western blotting

For western blotting, tissues were homogenized in 10× RIPA buffer (50 mM Tris-HCl pH 7.4; 5 mM EDTA; 150 mM NaCl; 1 mM Na-pyrophosphate; 1 mM NaF; 1 mM Na-vanadate; 1% NP-40) supplemented with cOmplete Mini Protease Inhibitor Cocktail Tablets (Roche) using a TissueLyser (QIAGEN) at 20 Hz for 2× 3 min. Samples were centrifuged (10 min, 13,000 rpm) and the supernatant without the upper fat layer was taken with a syringe. Protein determination was performed using the BCA method. 20 µg of total protein was separated on 10% SDS-acrylamid Tris-glycine gels and for western blotting, transferred to nitrocellulose membranes (GE) in a wet blotting system. After blocking (1 h) in 5% milk in TBS-T (20 mM Tris, 150 mM NaCl, 0.1% (v/v) Tween 20), the membranes were incubated with the primary antibodies overnight at 4 °C (diluted in 5% BSA in TBS-T). Following primary antibodies were used: rabbit-anti-Akt (1:1000, 9272, Cell Signaling),

rabbit-anti γ-Tubulin (1:2000, ab179503, abcam), rabbit-anti-pHSL (1:1000, 4139, Cell Signaling), rabbit-anti-HSL (1:1000; 4107, Cell Signaling), rat-anti-MAC2 (1:2000, sc-23938, Santa Cruz), rabbit-anti tyrosine hydroxylase (1:5000, ab137869, abcam), rabbit-anti-PLIN1 (1:1000, 9349, Cell Signaling), mouse-anti-UCP1 (1:2500; MAB6158, R&D). Incubation with respective secondary horseradish peroxidase-coupled antibodies was performed after washing with TBS-T. The membranes were developed with enhanced chemiluminescence (ECL) by using Amersham Imager 600 (GE). Signal quantification was performed using Li-Cor ImageStudioLite.

## Indirect calorimetry

For indirect calorimetry, mice were acclimated to metabolic cages (Promethion®, Sable Systems) at 22 °C for at least 2 days. Oxygen consumption, carbon dioxide production, food and water intake as well as locomotor activity were monitored continuously. The data files were analyzed according to the manufacturer (Sable Systems) using the Macro interpreter software. RQ was calculated as $V(CO_2)/V(O_2)$. To monitor body core temperature, telemetric transponders were implanted (G2 E-Mitter, Starr Life Sciences).

## Gene expression analysis of murine and human samples

Gene expression analysis by qPCR was performed as described (Sass et al, 2021). In brief, RNA was isolated from cell samples or whole tissues using the NucleoSpin RNA II kit (Macherey and Nagel) and transcribed into cDNA using High-Capacity cDNA Reverse Transcription kit (Applied Biosystems) according to the manufacturer's protocol. qPCR was performed using either Taq-Man® Assays or SYBR primer (see "Reagents and Tools table"). Expression levels were normalized to $Tbp$ or $36b4$ using the $2^{-\Delta\Delta Ct}$ method as indicated.

For mRNA sequencing (RNAseq), total RNA was sent to Novogene (UK) for quality check, library construction and transcriptome sequencing of RNA samples on a NovaSeq 6000 PE150 platform. Bioinformatics analysis included mapping to the mouse reference genome, gene expression quantification, differential expression analysis as well as GO enrichment analysis of differentially expressed genes. Genes with adjusted $P$ value < 0.01 and |log2(FoldChange)| > 0 were considered as differentially expressed. Classification of GO databases included three main branches (cellular component, molecular function, biological process) and GO terms with padj < 0.05 were considered significant enrichment.

## Phenome-wide association studies

For studying the association between protein-coding genes of P2x receptors and metabolic phenotypes in approximately 400,000 individuals from the UK biobank, we used the gene-based phenome-wide association study browser ExPheWas (https://exphewas.statgen.org/v1) (Legault et al, 2022).

## Flow cytometric analysis

Flow cytometric analysis to distinguish tissue-resident from vascular immune cells was performed as previously described (Rissiek et al, 2014) and adapted for BAT. In brief, anesthetized

mice received 2 µg of anti-CD45-Per-CP antibody (clone 30-F11, 103130, Biolegend) to stain cells present in blood vessels (Schwarz et al, 2020). After mice were transcardially perfused with PBS, BAT (tissue of two mice was pooled) was minced and digested in PBS containing 10 mM $CaCl_2$, 1.5 units/ml Collagenase D and 2.4 units/ml dispase II for 45 min at 37 °C. The homogenate was filtered through a 100 µm cell strainer. Afterwards, the stromal vascular fraction (SVF) was isolated by centrifugation (5 min at $600 \times g$, 4 °C). The pellet was re-suspended in FACS buffer containing 0.1% bovine serum albumin (Sigma) and centrifuged at 1600 rpm for 5 min. To lyse erythrocytes, the cell pellets were incubated with lysis buffer (155 mM $NH_4Cl$, 10 mM $KHCO_3$, 0.1 mM EDTA, pH 7.2) for 3 min at room temperature. After a washing step with FACS buffer, the cells were incubated in FACS buffer containing anti-mouse CD16/CD32 (1:100 in FACS buffer, 2.4G2, BE0307, BioXCell) to block unspecific binding of fluorochrome-conjugated antibodies.

For detection of cell bound anti-P2X7 Nanobodies, cells were washed with FACS buffer, centrifuged and re-suspended cells were incubated with anti-VHH-msIgG1 (clone mab77) for 20 min at 4 °C. Cells were then washed with FACS buffer and stained with antibodies against immune cell surface markers and anti-msIgG1-Bv421 (clone RMG1-1, 406615, BioLegend). For immune cell phenotyping, the following antibodies-fluorochrome conjugates were used: anti-CD45-PE-Cy7 (clone 30-F11, 103114, BioLegend), anti-CD11b-BUV737 (clone M1/70, 612800, BD Bioscience), anti-CD206-APC(clone C068C2, 141707, Biolegend), anti-Ly6G-AF700 (clone 1A8, 127621, Biolegend), anti-CD8a-BUV395 (clone OX-8, 740257, BD Bioscience) anti-F4/80-BUV563 (clone T45-2342, 749284, BD Bioscience), anti-CD4-Bv421 (clone GK1.5, 100437, Biolegend), anti-CD3e-Bv650 (clone 500A2, 740461, BD Bioscience), anti-NK1.1-Bv711 (clone PK136, 108745, Biolegend), anti-B220-PE (clone RA3-6B2, 103207, Biolegend), anti-CD11c-PE-Dazzle (clone N418, 117347, Biolegend) and LIVE/DEAD® Fixable Near-IR (L10119, ThermoFisher) for 20 min at 4 °C.

For evaluation of P2X4 expression on peritoneal macrophages and mast cells, mice were sacrificed and subjected to peritoneal lavage using 5 ml ice cold PBS + EDTA (2 mM, Gibco). Macrophages and mast cells were distinguished by the expression of CD11b and FceR1 using anti-CD11b-Bv421 (clone M1/70, 101235, Biolegend) and anti-FceR1-PE (clone MAR-1, 12-5898-82, ThermoFisher). Expression of P2X4 was detected by anti-P2X4-AF647 (clone RG96A246, UKE inhouse production, (Bergmann et al, 2019)).

For detection of activated Caspase-1 in macrophages FAM-FLICA(R) Caspase 1 Assay Kit (Immunochemistry Technologies) was according to manufacturer's instruction. Freshly prepared FAM-FLICA solution was added to the phenotyping mAb cocktail and was allowed to penetrate cells and bind to activated Caspase-1 for 30 min at 4 °C.

After staining, cells were washed twice with FACS buffer, re-suspended in 100 µl FACS buffer and analyzed by flow cytometry using BD FACS Celesta (Beckton Dickinson) or BD FACS Symphony A3 (Beckton Dickinson). FACS data analysis was performed using BD FlowJo 10.8.1.

## Lipidomics

Plasma and BAT samples were prepared and analyzed according to the Lipidyzer Platform protocols using the corresponding standard kits (SCIEX). In short, 50 µL of plasma or 5 mg of snap-frozen homogenized BAT were extracted by an adjusted MTBE/methanol extraction (Matyash et al, 2008). Samples were mixed with the internal standards for the Lipidyzer Platform (SCIEX) during lipid extraction. After concentration, lipid extracts were reconstituted in running buffer (10 mM ammonium acetate, dichloromethane (50): methanol (50)). For quantitative measurement of lipid species, extracts were directly infused into an ESI-QqQ system run in multiple reaction monitoring (MRM) mode (QTRAP 5500; SCIEX)). Data analysis and lipid quantification was achieved in an automated manner using the Lipidyzer software (Lipidomics Workflow Manager software; SCIEX).

## Statistical methods

Statistical analysis was performed using GraphPad Prism 9.2. Data are expressed as mean ± S.E.M. Two-sided Student's $t$ test was used to compare two groups. For more than two groups two-way ANOVA followed by Sidak's or Tukey's correction for multiple comparison was performed. No method was used to determine whether the data met assumptions of either Student's $t$ test or ANOVA. The statistical parameters (i.e., $P$ values, numbers of biological repeats) can be found in the figure legends. Sample size was calculated using G*power. In accordance with 3Rs principles, we used a sample size that was able to detect a significant change of 20% with $P < 0.05$. The animals were randomised according to sex, age and body weight. Allocations of animals was performed in a randomized manner. Experiments were not blinded. No exclusion or inclusion criteria were used for data analyses.

## Data availability

This study includes no data deposited in external repositories.

The source data of this paper are collected in the following database record: biostudies:S-SCDT-10_1038-S44319-025-00642-y.

## Peer review information

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

## Acknowledgements

JH, AW, AP, TM, BR and FK-N were supported by the Deutsche Forschungsgemeinschaft-funded research consortium SFB1328, project-ID: 335447727. JH, LS, AW and AP were supported by the Deutsche

Forschungsgemeinschaft-funded research consortium SFB-Transregio 333, project-ID: 450149205. The authors thank Sandra Ehret, Birgit Henkel and Paul Pertzborn for excellent technical assistance. We also acknowledge Natasa Petrovic (Wenner Gren Institute, Stockholm, Sweden) for providing RNAseq data from mice housed in a humanized environment. We thank Wenfei Sun and Lorenz Adlung for reanalyzing the murine and human snRNAseq data.

## Author contributions

**Michelle Y Jaeckstein**: Conceptualization; Data curation; Formal analysis; Investigation; Visualization; Methodology; Writing—original draft. **Alexander W Fischer**: Conceptualization; Data curation; Investigation; Methodology; Writing—review and editing. **Björn Rissiek**: Data curation; Funding acquisition; Investigation; Visualization; Methodology; Writing—review and editing. **Tobias Staehler**: Investigation; Methodology; Writing—review and editing. **Markus Heine**: Validation; Investigation; Methodology; Writing—review and editing. **Janina Behrens**: Validation; Investigation; Methodology; Writing—review and editing. **Oliver Mann**: Resources; Methodology; Writing—review and editing. **Alexander Pfeifer**: Resources; Funding acquisition; Methodology; Writing—review and editing. **Tim Magnus**: Resources; Supervision; Funding acquisition; Writing—review and editing. **Christian Schlein**: Conceptualization; Funding acquisition; Investigation; Methodology; Writing—review and editing. **Anna Worthmann**: Funding acquisition; Investigation; Methodology; Writing—review and editing. **Ludger Scheja**: Funding acquisition; Investigation; Methodology; Writing—original draft; Writing—review and editing. **Friedrich Koch-Nolte**: Conceptualization; Resources; Supervision; Funding acquisition; Investigation; Methodology; Writing—review and editing. **Joerg Heeren**: Conceptualization; Data curation; Supervision; Funding acquisition; Investigation; Visualization; Writing—original draft; Project administration; Writing—review and editing.

Source data underlying figure panels in this paper may have individual authorship assigned. Where available, figure panel/source data authorship is listed in the following database record: biostudies:S-SCDT-10_1038-S44319-025-00642-y.

## Funding

## Disclosure and competing interests statement

The authors declare no competing interests.

# Expanded View Figures

**Figure EV1.   Related to Fig. 1. Inflammatory BAT remodeling is triggered by imbalanced BAT activation.**

(**A–G**) Male *Ucp1$^{-/-}$* mice and wild-type (WT) littermates were housed at 22 °C. (**A**) iBAT weight ($n = 5$). (*$P = 0.0004$). (**B**) BAT gene expression of *Ucp1*, fibrosis and inflammatory marker genes ($n = 5$). (*Ucp1*: *$P = 2.0040E\text{-}05$; *Tnf*: *$P = 0.0054$; *Ccl2*: *$P = 0.0275$; *Il1b*: *$P = 0.0703$; *Ifng*: *$P = 0.0117$; *Emr1*: *$P = 0.0044$; *Cd4*: *$P = 0.0064$; *Cd8b1*: *$P = 0.0055$; *Timp1*: *$P = 0.0172$; *Col1a1*: *$P = 0.0091$; left to right). (**C**) Body weight ($n = 5$). (**D**) BAT images of HE and Sirius Red stainings as well as immunostaining of MAC2. Scale bar, 50 μm. ($n = 4$). For better comparison, images of all individual mice are presented here, which includes the representative image shown in the main Fig. 1. (**E**) Quantification of lipid droplet area as shown in (**D**) ($n = 4$). (**F**) Quantification of MAC2 staining as shown in (**D**) ($n = 4$). (*$P = 0.0038$). (**G**) Quantification of Sirius red staining as shown in (**D**) ($n = 4$). (*$P = 0.0077$). (**H**) Male *Ucp1$^{-/-}$* mice and wild-type (WT) littermates were housed at 22 °C or 6 °C. Western Blot of BAT ($n = 5$). (**I**) Quantification of Western blot shown in (**H**). (UCP1: *$P = 0.0032$; #$P = 0.0016$; *$P = <0.0001$; MAC2: *$P = 0.0136$; *$P = 0.0003$; left to right). (**J**) Quantification of Western blot shown in Fig. 1C. (UCP1: *$P = <0.0001$; #$P = <0.0001$; *$P = <0.0001$; MAC2: *$P = 0.0001$; #$P = 0.0005$; TH: *$P = 0.0037$; #$P = <0.0001$; *$P = <0.0001$; left to right). Data are presented as mean values ± SEM. *$P < 0.05$ by Student's *t* test comparing WT vs. *Ucp1$^{-/-}$*-mice (**A–C, E–G**) or two-way ANOVA comparing WT vs. *Ucp1$^{-/-}$*-mice (**I, J**). #$P < 0.05$ by two-way ANOVA comparing 22 °C vs 6 °C (**I**) or sham vs DNV (**J**). *N* values indicate biological replicates.

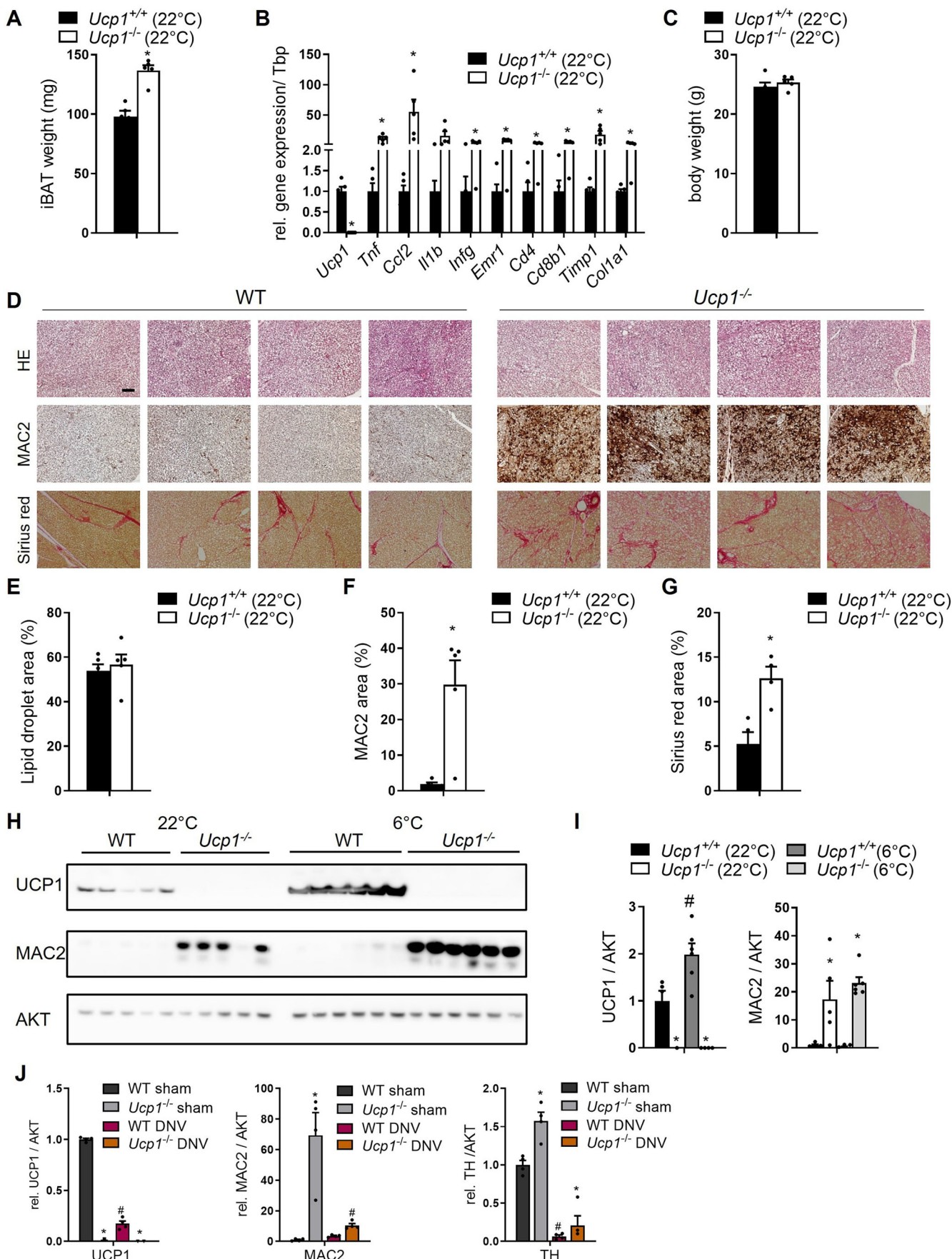

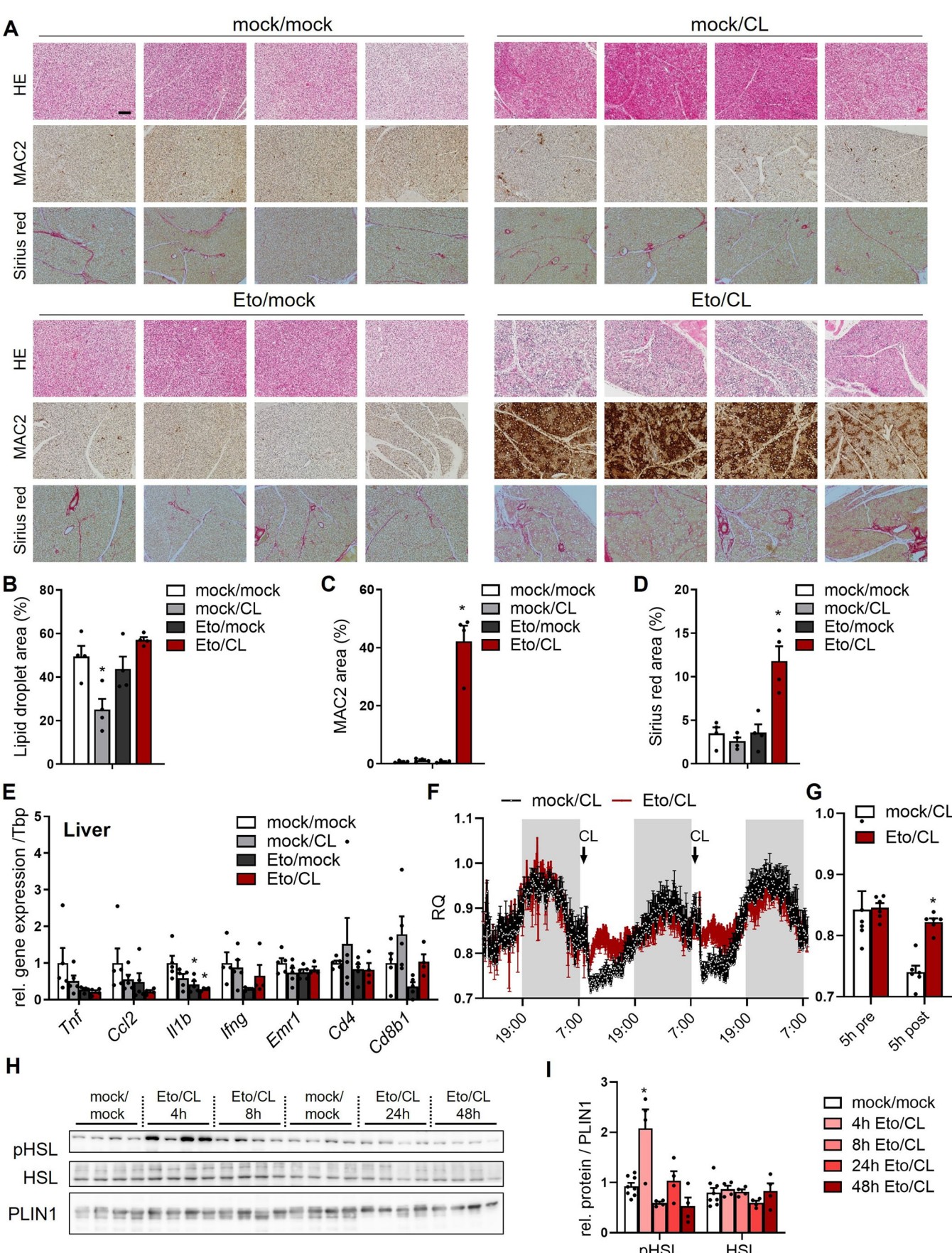

◄ **Figure EV2.  Related to Fig. 1. Inflammatory BAT remodeling is triggered by imbalanced BAT activation.**

Wild-type mice housed at 22 °C were injected daily for three subsequent days with either vehicle (mock/mock), with the beta-3-adrenergic agonist CL316,243 alone (mock/CL), with the inhibitor of beta oxidation etomoxir alone (Eto/mock) or with the combination of both (Eto/CL). (A) Representative images of BAT of individual mice employing HE, Sirius Red and MAC2 (immune)-stainings, respectively. Scale bar, 50 μm. ($n = 4$). For better comparison, images of all individual mice are presented here, which includes the representative image shown in the main Fig. 1. (B) Quantification of lipid droplet area ($n = 4$, 3 images per section per mouse). (*$P = 0.0073$). (C) Quantification of MAC2 staining as shown in (A) ($n = 4$, 3 images per section per mouse). (*$P = <0.0001$). (D) Quantification of Sirius red staining as shown in (A) ($n = 4$, 3 images per section per mouse). (*$P = 0.0004$). (E) Liver gene expression of inflammatory marker genes ($n = 4$–5). (*$P = 0.0246$; *$P = 0.0095$; left to right). (F) Respiratory quotient (RQ) in mock/CL and Eto/CL mice. The first and second CL injection are indicated by arrows ($n = 6$). (G) Quantification of respiratory quotient for 5 h pre and post CL injection ($n = 6$). (*$P = 4.6982E-05$). (H) Wild-type mice were treated with Eto and CL for indicated time spans. Western Blot of Plin1, HSL and phospho HSL (pHSL) in WAT samples ($n = 4$). (I) Quantification of Western blot shown in (H). (*$P = 0.0004$). Data are presented as mean values ± SEM. *$P < 0.05$ by ANOVA (B-E, I) compared to mock control or Student's *t* test (G). *N* values indicate biological replicates.

**A**

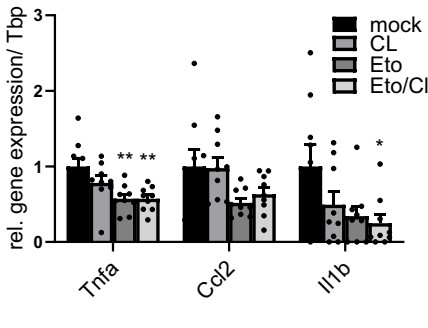

**B murine snRNAseq data**

Behrens et al., bioRxiv  https://doi.org/10.1101/2025.03.28.646056)

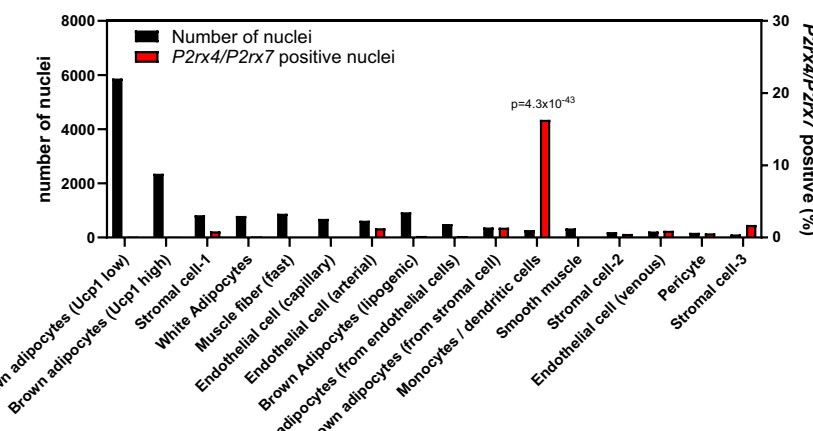

**C**

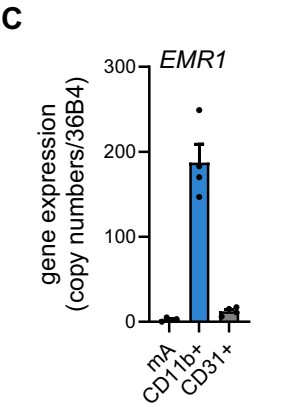

**D**

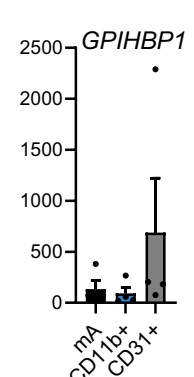

**E**

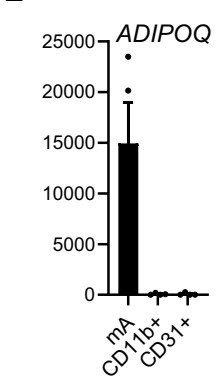

**F**

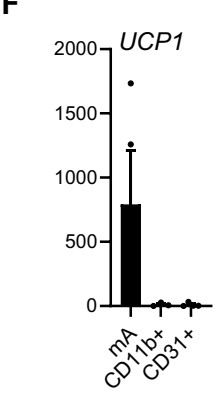

**G human snRNAseq data**

Sun et al., Nature 2020

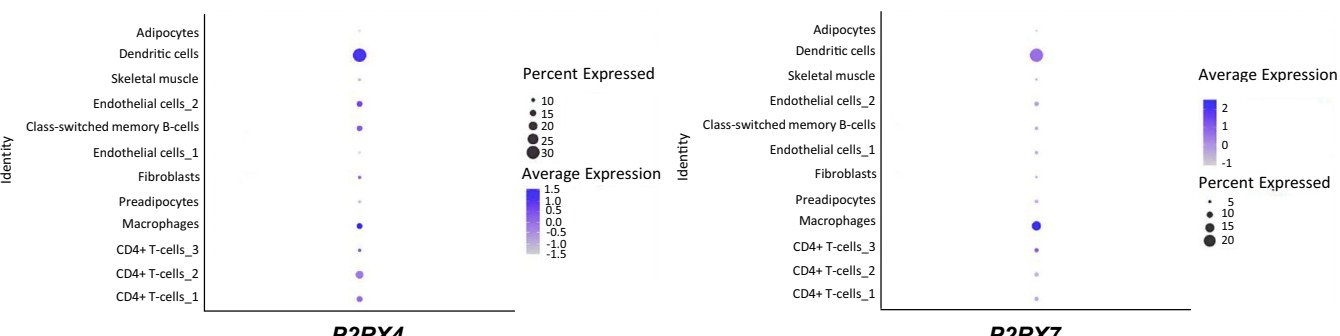

◀  **Figure EV3.    Related to Fig. 3. Brown adipocyte dysfunction causes ATP secretion and is linked to purinergic receptor signaling in BAT.**

(A) Brown adipocytes differentiated from BAT stromal vascular fractions were treated for 24 h with or without Eto (25 μM) in the presence or absence of CL (50 nM). Gene expression of cytokines ($n = 9$). (*$P = 0.0053$; *$P = 0.0054$; *$P = 0.0393$, left to right). (B) Analysis of published murine snRNAseq data (Behrens et al, Mol Metab. 2025 Nov;101:102252. doi: 10.1016/j.molmet.2025.102252.). Presentation of *P2rx4/P2rx7*-double positive nuclei in a dataset of murine single nucleus RNA sequencing. Cell types with a nuclei count above 100 were re-analyzed. (C–F) Human BAT samples from four donors were sorted using low speed centrifugation to enrich mature adipocytes (mA), followed by MACS® to isolate CD31+ endothelial cells and CD11b+ myeloid cells. (C) Gene expression of the macrophage marker *EMR1* in the three fractions ($n = 4$). (D) Gene expression of the endothelial cell marker *GPIHBP1* in the three fractions ($n = 4$). (E) Gene expression of the adipocyte marker *ADIPOQ* in the three fractions ($n = 4$). (F) Gene expression of the thermogenic brown adipocyte marker *UCP1* in the three fractions ($n = 4$). (G) Analysis of published human snRNAseq data (Sun et al, Nature 2020). Expression of *P2RX4/P2RX7* in various cell clusters of human brown adipose tissue. Data are presented as mean values ± SEM. *$P < 0.05$ by ANOVA compared to mock control (A, C–F). *N* values indicate biological replicates.

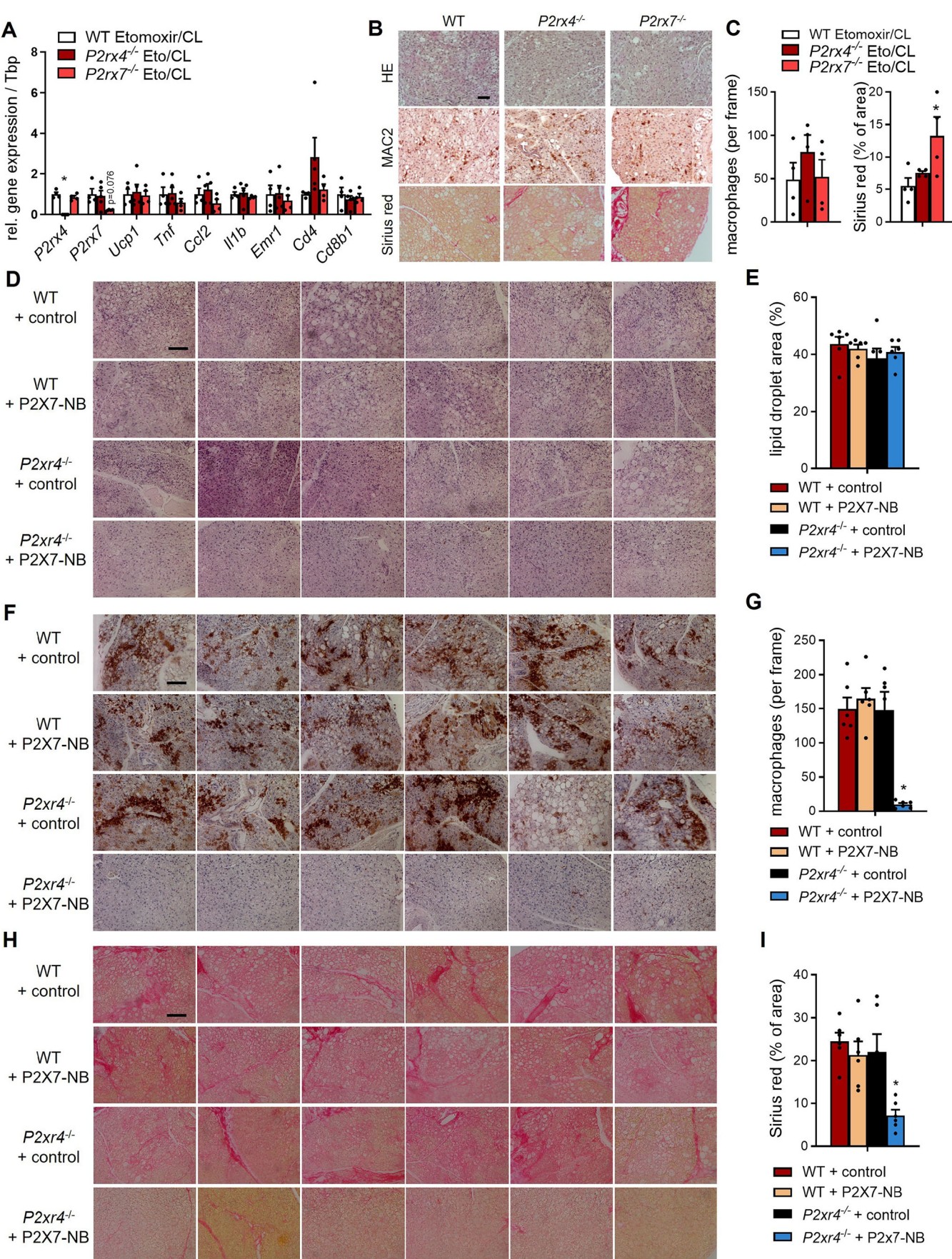

◀ **Figure EV4. Related to Fig. 4. Combined inhibition of P2X4 and P2X7 prevents pharmacologically induced BAT degeneration thereby preserving thermogenic capacity.**

(**A–C**) Wild-type (WT), *P2rx4*[-/-] and *P2rx7*[-/-] mice were daily injected with Eto and CL on two consecutive days. (**A**) BAT gene expression of *Ucp1*, purinergic receptors and inflammatory markers ($n = 4$–5). (*$P = $ <0.0001). (**B**) Representative BAT images of HE, Sirius Red and MAC2 (immune)-stainings. Scale bar, 50 μm. (**C**) Quantification of macrophages and Sirius red staining ($n = 4$, 3 images per section per mouse). (*$P = 0.0284$). (**D–I**) Wild-type (WT) and *P2rx4*[-/-] mice were pretreated with the P2X7-inhibiting nanobody or vehicle (control). Then, mice were daily injected with Eto and CL on two consecutive days. (**D**) BAT images of HE staining of individual mice. Scale bar, 50 μm. For better comparison, images of all individual mice are presented here, which includes the representative image shown in the main Fig. 4. (**E**) Quantification of lipid droplet area ($n = 6$, 3 images per section per mouse). (**F**) BAT images of MAC2 immunestaining of individual mice. Scale bar, 50 μm. For better comparison, images of all individual mice are presented here, which includes the representative image shown in the main Fig. 4. (**G**) Quantification of macrophages ($n = 6$, 3 images per section per mouse). (*$P = $ <0.0001). (**H**) BAT images of Sirius Red staining of individual mouse. Scale bar, 50 μm. For better comparison, images of all individual mice are presented here, which includes the representative image shown in the main Fig. 4. (**I**) Quantification of Sirius Red staining ($n = 6$, 3 images per section per mouse). (*$P = 0.0012$). Data are presented as mean values ± SEM. *$P < 0.05$ by ANOVA compared to WT Eto/CL (**A**, **C**) or WT + control (**E**, **G**, **I**). *N* values indicate biological replicates.

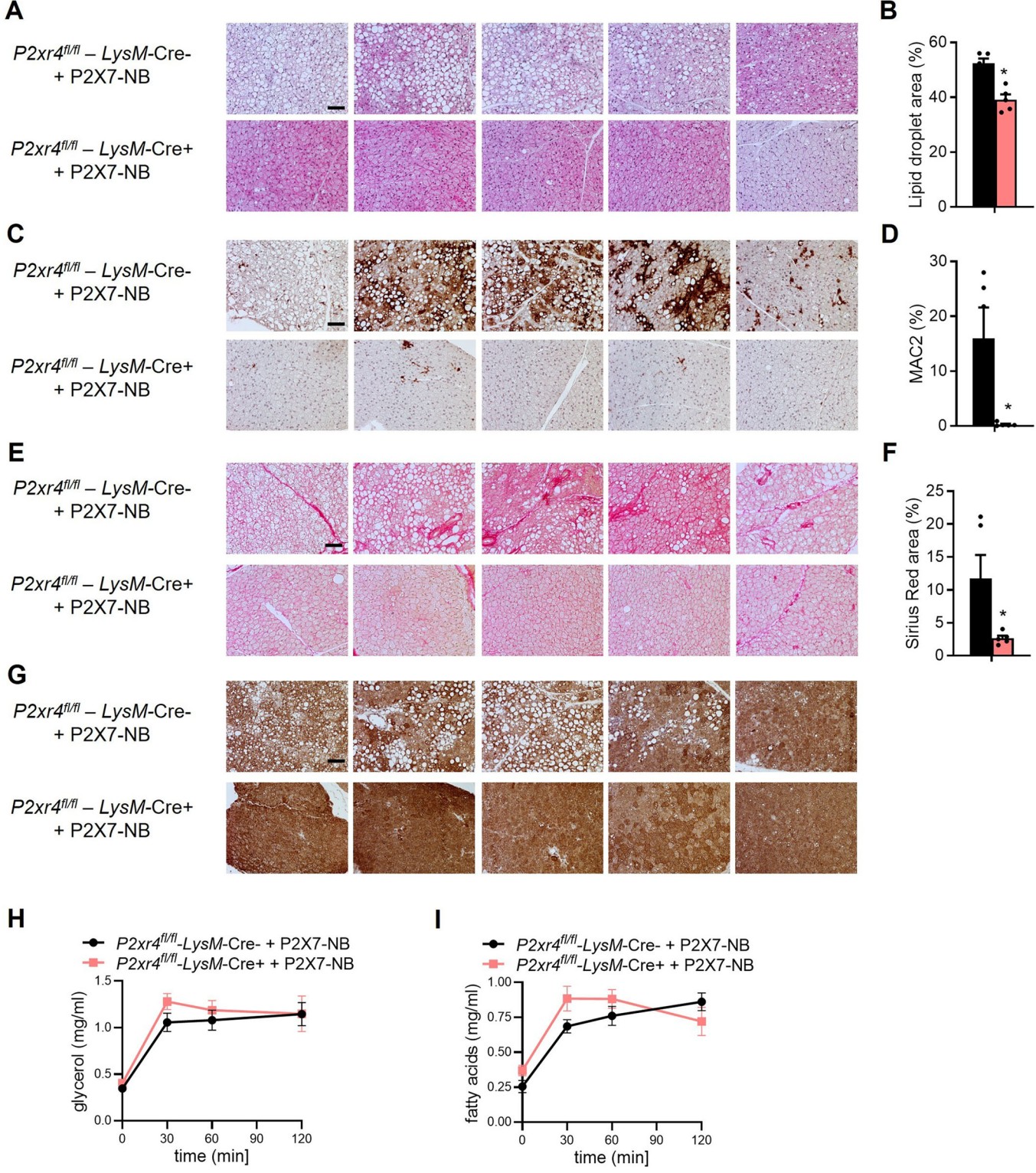

◄   **Figure EV5.   Related to Fig. 5. Myeloid P2X4 expression determines BAT degeneration.**

(A–G) *P2rx4*<sup>fl/fl</sup>-LysM<sup>Cre-</sup> and *P2rx4*<sup>fl/fl</sup>-LysM<sup>Cre+</sup> mice were pretreated with the P2X7-inhibiting nanobody. Then, mice were daily injected with Eto and CL on two consecutive days. (A) BAT images of HE staining of individual mice. Scale bar, 50 μm. For better comparison, images of all individual mice are presented here, which includes the representative image shown in the main Fig. 5. (B) Quantification of lipid droplet area ($n = 5$, 3 images per section per mouse). (*$P = 0.0008$). (C) BAT images of MAC2 immunestaining of individual mice. Scale bar, 50 μm. For better comparison, images of all individual mice are presented here, which includes the representative image shown in the main Fig. 5. (D) Quantification of macrophages ($n = 5$, 3 images per section per mouse). (*$P = 0.0239$). (E) BAT images of Sirius Red staining of individual mice. Scale bar, 50 μm. For better comparison, images of all individual mice are presented here, which includes the representative image shown in the main Fig. 5. (F) Quantification of Sirius Red staining ($n = 5$, 3 images per section per mouse). (*$P = 0.0358$). (G) BAT images of UCP1 immunestaining of individual mice. Scale bar, 50 μm. For better comparison, images of all individual mice are presented here, which includes the representative image shown in the main Fig. 5. (H) + I *P2rx4*<sup>fl/fl</sup>-LysM<sup>Cre-</sup> and *P2rx4*<sup>fl/fl</sup>-LysM<sup>Cre+</sup> mice were pretreated with the P2X7-inhibiting nanobody. Then, mice were injected with CL. (H) Plasma glycerol levels at different time points after CL injection ($n = 6$). (I) Plasma fatty acid levels at different time points after CL injection ($n = 6$). Data are presented as mean values ± SEM. *$P < 0.05$ by Student's *t* test. *N* values indicate biological replicates.

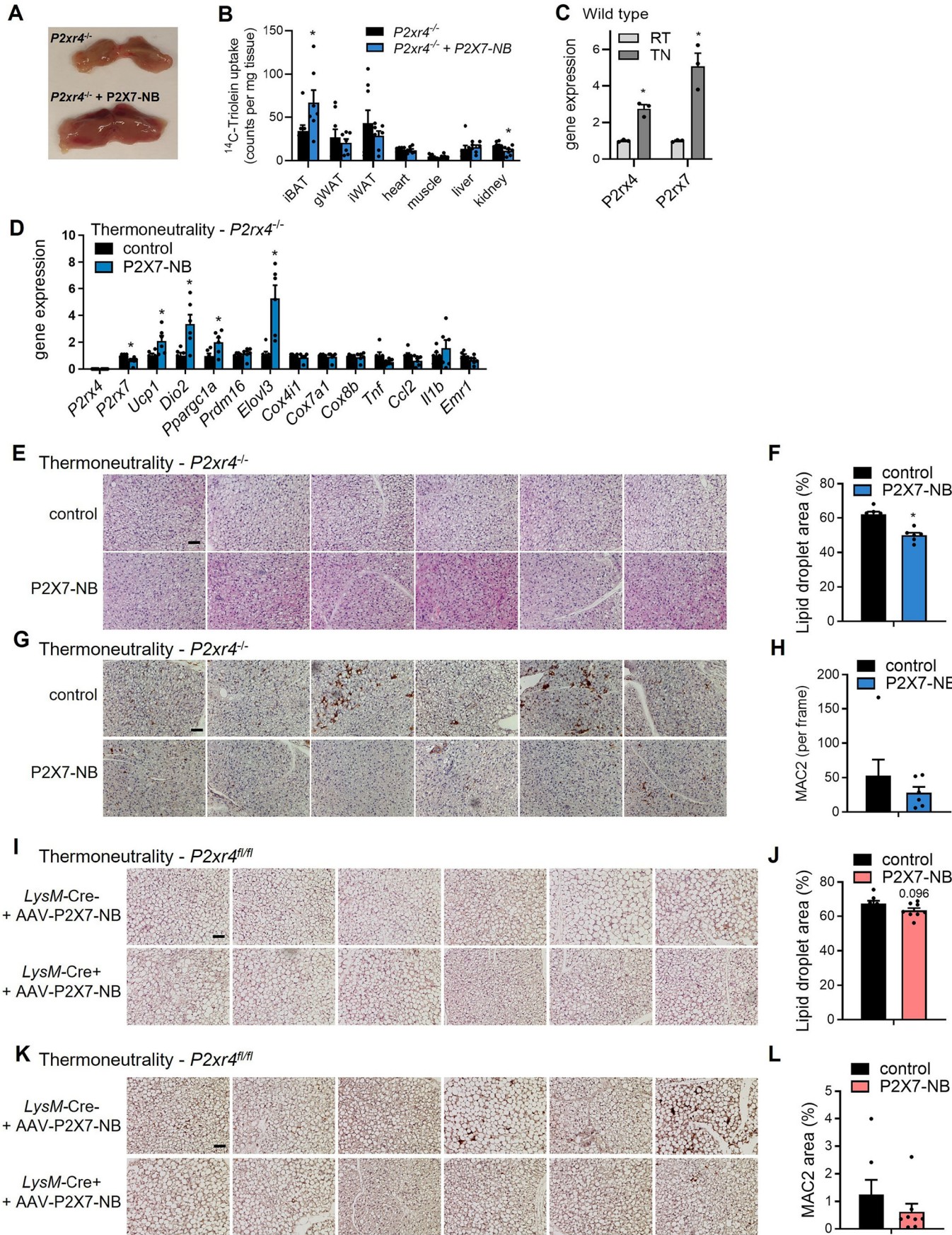

**Figure EV6. Related to Fig. 6. P2X4/P2X7 blockade improves systemic glucose and lipid metabolism in distinct models of BAT degeneration.**

(A, B) $P2rx4^{-/-}$ mice were pretreated with the P2X7-inhibiting nanobody or vehicle. Then, mice were daily injected with Eto and CL on two consecutive days. (A) Images of BAT after two days of Eto and CL treatment. (B) Uptake of i.v. injected TRL labeled with $^{14}$C-triolein into various organs per mg tissue ($n = 7$–8). (*p = 0.0469; *p = 0.0265; left to right). (C) Gene expression of $P2rx4$ and $P2rx7$ in wild-type mice housed at room temperature (RT) or thermoneutrality (TN). ($P2rx4$: *p = 0.0015; $P2rx7$: *$P$ = 0.0045; left to right). (D–H) $P2rx4^{-/-}$ mice receiving P2X7-inhibiting nanobody or vehicle (control) were housed at thermoneutrality (30 °C) for 2 weeks. ($P2rx7$: *$P$ = 0.0068; $Ucp1$: *$P$ = 0.0238; $Dio2$: *$P$ = 0.0095; $Ppargc1a$: *$P$ = 0.0263; $Elovl3$: *$P$ = 0.0022; left to right). (D) Gene expression of purinergic receptors, thermogenic and inflammatory marker genes ($n = 6$). (E) BAT images of HE staining of individual mice. Scale bar, 50 μm. For better comparison, images of all individual mice are presented here, which includes the representative image shown in the main Fig. 6. (F) Quantification of lipid droplet area ($n = 6$, 3 images per section per mouse). (*$P$ = 0.0003). (G) BAT images of MAC2 immunestaining of individual mice. Scale bar, 50 μm. For better comparison, images of all individual mice are presented here, which includes the representative image shown in the main Fig. 6. (H) Quantification of macrophages ($n = 6$, 3 images per section per mouse). (I–L) $P2rx4^{fl/fl}$-LysM$^{Cre-}$ and $P2rx4^{fl/fl}$-LysM$^{Cre+}$ mice received an AAV encoding for the P2X7-inhibiting nanobody (AAV-P2X7-NB) or control vector. Two weeks after infection, mice were housed at thermoneutrality (30 °C) for 2 weeks. (I) BAT images of HE staining of individual mice. Scale bar, 50 μm. For better comparison, images of all individual mice are presented here, which includes the representative image shown in the main Fig. 6. (J) Quantification of lipid droplet area ($n = 6$, 3 images per section per mouse). (K) BAT images of MAC2 immunestaining individual mice. Scale bar, 50 μm. For better comparison, images of all individual mice are presented here, which includes the representative image shown in the main Fig. 6. (L) Quantification of macrophages ($n = 6$, 3 images per section per mouse). Data are presented as mean values ± SEM. *$P < 0.05$ by Student's $t$ test. $N$ values indicate biological replicates.

