## [Peer Review File · EMBO Reports]

Purinergic adipocyte-macrophage crosstalk promotes degeneration of thermogenic brown adipose tissue

Michelle Jaeckstein, Alexander Fischer, Björn Rissiek, Tobias Staehler, Markus Heine, Janina Behrens, Oliver Mann, Alexander Pfeifer, Tim Magnus, Christian Schlein, Anna Worthmann, Ludger Scheja, Friedrich Koch-Nolte, and Joerg Heeren

Corresponding author(s): Joerg Heeren (heeren@uke.de)

Review Timeline:	Transfer Date:	23rd Sep 25
	Editorial Decision:	26th Sep 25
	Revision Received:	21st Oct 25
	Accepted:	3rd Nov 25

Editor: Esther Schnapp

Transaction Report: This manuscript was transferred to EMBO reports following peer review at The EMBO Journal.

Referee #1

As former reviewer #4 of the original submission to another journal, my comments below provide updates on the feedback I gave previously. I will not repeat earlier remarks.

All of the strengths noted before remain: the study is a mechanistic tour de force; the major new finding is the identification of a key role for P2RX4/P2RX7 in "BAT degeneration" and HFD/TNZ (potentially also relevant to long-term involution or de-differentiation associated with aging); the experiments are well-designed and executed; and the overall rigor is high.

However, several weaknesses raised earlier were unfortunately not adequately addressed, and they diminish the impact of what could otherwise be a high-quality publication:

1. Unclear focus of the manuscript: Roughly three-quarters of the Results section centers on an artificial pharmacological induction of BAT "degeneration" using CL+Eto, largely in UCP1-/- mice. While the rationale for this model is now more clearly stated-that it provides a simple, rapid approach to induce BAT "degeneration" and thus identify potential mechanisms or drug targets-devoting so much space to characterizing the model detracts from the physiologically relevant sections, which appear only toward the end. Importantly, the final section does not include a comprehensive enough characterization of P2X4/7 function in wild-type mice under HFD, thermoneutrality (TNZ), or aging conditions. In my view, the manuscript does not strike an appropriate balance between these two aspects, leading to confusing and weakly justified statements such as: "the pharmacological treatment regimen based on etomoxir and CL caused inflammatory BAT degeneration, mimicking the phenotype of UCP1-deficient mice." I do not believe mimicking the UCP1-deficient phenotype is of broad interest-unless this were intended as a methods paper, which it is not.

2. Mechanism of ATP secretion remains insufficiently addressed: The role of Pannexin 1 in ATP secretion by brown adipocytes is investigated only in vitro, in the absence of nerve terminals or other relevant cell types present in vivo. This is a crucial limitation: without in vivo validation, the conclusions remain only suggestive and cannot establish the relative contributions of pannexin1 in brown adipocytes versus VNUT-mediated ATP release from other cell types, including adipocytes.

3. Unresolved link between myeloid P2X4/P2X7 and brown adipocyte dysfunction: The new and revised data help clarify the impact of myeloid-expressed P2X4/7 on BAT phenotype.

However, the mechanism by which this leads to brown adipocyte dysfunction and BAT degeneration remains unclear. This represents a major gap in what is otherwise a strong mechanistic study.

4. Overemphasis on "BAT degeneration": While the revised framing around "BAT degeneration" is better supported by the data, it also detracts from the physiological significance of the work. A key unresolved question is whether the effects of CL+Eto are reversible-and if so, whether reversibility occurs within a timeframe and through mechanisms consistent with de-differentiated BAT in obese or TNZ conditions, where BAT can be reactivated by CL, ATP γ S, cold exposure, or other stimuli.

5. The role of additional ATP-activated purinergic receptors cannot be disregarded merely because their expression levels appear low in the current assays. Given the opposing roles of P2X5 versus P2X4/7 in BAT function, it is likely that interactions among these receptors-and potentially other purinergic receptors-play an important in vivo role that warrants consideration.

6. Relying solely on transcriptomics is inadequate to exclude the possibility that CL/Eto exerts BAT-independent effects that indirectly impact BAT function and degeneration.

Referee #2

In this revised manuscript, the authors have provided additional data to address the reviewers' concerns, along with further clarifications and arguments regarding aspects of the methodology and data interpretation. Overall, the revision strengthens and clarifies the manuscript; however, a few minor updates are still advisable:

1. The authors responded to the shared concern regarding the rationale for focusing on macrophage-adipocyte crosstalk by emphasizing that analyses across several selected immune cell types were performed. While this is a reasonable argument, the approach remains somewhat biased and incomplete. In the current era of single-cell analyses, unbiased approaches that examine all cellular components within adipose tissue and systematically map intercellular communications are both technically feasible and methodologically standardized. It is therefore advisable that the authors acknowledge this limitation in their approach.

2. The physiological relevance of the Eto/CL model remains debatable. Nevertheless, the

authors' arguments highlighting the similarity and relevance of this model to physiological settings such as thermoneutrality are appreciated. It would, however, be advisable for the wording in the manuscript to explicitly acknowledge the limitations of this model.

3. Most importantly, the lack of cell type-specific knockout models remains a significant limitation of the current study. The authors argue that the roles of P2X4 and P2X7 in the myeloid lineage have been validated, citing their restricted expression patterns based on scRNA-seq data, and propose that combining a P2X4 myeloid-specific knockout with a P2X7 antagonist could approximate a myeloid-specific double knockout. However, the absence of detectable expression in other cell types based on scRNA-seq data requires cautious interpretation, given that scRNA-seq often has low sequencing depth and may underestimate low-abundance transcripts, including receptor genes. It is therefore advisable that the authors acknowledge the methodological limitations of their genetic models.

Dear Joerg,

Thank you for the transfer of your manuscript with referee reports to EMBO reports. As discussed, we would like to publish your study, however, all final referee concerns will need to be addressed in the ms text. Please send us a detailed point-by-point response to the final referee comments as well as to the editorial requests that will also need to be addressed:

- Please submit the ms file as a word file.
- The Data Availability Statement (DAS) is missing: Please add a data availability section to the end of the methods providing access to data deposited in public databases. If you have not deposited such data, please add a sentence to the DAS that explains that.
- The conflict of interest subheading needs to be renamed to Disclosure and Competing Interests Statement.
- The author credits needs to be removed from the ms file. All credits need to be entered during online ms submission.
- Please co-submit a completed author checklist, which you can download from our author guidelines <<https://www.embopress.org/page/journal/14693178/authorguide>>. The completed author checklist will also be part of the transparent peer-review file.
- All funding information also needs to be entered in our online ms submission system. Please do so with your next submission.
- All main and EV figures need to be uploaded as individual, high quality figure files. You can find more information in our guide to authors online.
- A callout for Fig. EV7 is missing, please add. All EV figures need to be uploaded as individual figure files and their legends need to be placed after the main figure legends.
- The 5 EV tables uploaded all should be called Datasets, except Table EV4 that I think should be part of the Reagents and Tools table, or it could be a Table EV1. The datasets need to be renamed to Dataset EV1-EV4; and the callouts in the ms need to be updated too.
- The Reagents & Tools TABLE needs to be removed from the ms file and uploaded separately.
- Materials and Methods should be just Methods
- Our routine image analysis of to be accepted ms detected several figure panel reuses: between Figure 1 and Figure EV1, between Figure 1 and Figure EV2, between Figure 4 and Figure EV4, Figure 5 and EV5, Figure 6 and EV6. All these re-uses need to be explained and mentioned in the EV figure legends.

Figure Legends - Comments

- Please note that the exact p values are not provided in the legends of figures 1B, D, E, G, H, I, K, L; 2A-D; 3A-L, N; 4B, D, F, H, J, K, L; 5H-J; 6A-D, I, K, M, N, O, P; EV1 A, B, C, E, F, G, I, J; 2B, C, D, G, I; EV3 A, EV4 C, E, G, I; EV5 B, D, F; EV6 B, C, D, F. Please provide exact values as reasonable.
- Please indicate the statistical test used for data analysis in the legend of figure 2M.
- "degenerative" or "degeneration" should be deleted from the second sentence in the abstract.

EMBO press papers are accompanied online by A) a short (1-2 sentences) summary of the findings and their significance, B) 2-3 bullet points highlighting key results and C) a synopsis image that is exactly 550 pixels wide and 200-600 pixels high (the height is variable). The synopsis image should provide a sketch of the major findings, like a graphical abstract. Please note that text needs to be readable at the final size. Please send us this information along with the final manuscript.

I look forward to seeing a final version of your manuscript as soon as possible. Please use this link to submit your revision: <https://embor.msubmit.net/cgi-bin/main.plex>

Kind regards,
Esther

Point to point**Editorial requests EMBO Reports**

- The Data Availability Statement (DAS) is missing: Please add a data availability section to the end of the methods providing access to data deposited in public databases. If you have not deposited such data, please add a sentence to the DAS that explains that.

We added the DAS statement. The source data will be available at Biostudies, we do not have additional data that are deposited in public databases.

- The conflict of interest subheading needs to be renamed to Disclosure and Competing Interests Statement.

We renamed the section.

- The author credits needs to be removed from the ms file. All credits need to be entered during online ms submission.

We removed the author credits.

- Please co-submit a completed author checklist, which you can download from our author guidelines <<https://www.embopress.org/page/journal/14693178/authorguide>>. The completed author checklist will also be part of the transparent peer-review file.

We filled the author checklist.

- All funding information also needs to be entered in our online ms submission system. Please do so with your next submission.

Ok.

- All main and EV figures need to be uploaded as individual, high quality figure files. You can find more information in our guide to authors online.

We provided high quality figure files.

- A callout for Fig. EV7 is missing, please add. All EV figures need to be uploaded as individual figure files and their legends need to be placed after the main figure legends.

EV7 was excluded and is used as synopsis image. Therefore, there is no callout for Fig. EV7 necessary.

- The 5 EV tables uploaded all should be called Datasets, except Table EV4 that I think should be part of the Reagents and Tools table, or it could be a Table EV1. The datasets need to be renamed to Dataset EV1-EV4; and the callouts in the ms need to be updated too.

We renamed the EV tables to Datasets. As suggested, table EV4 is now part of the Reagents and Tools table.

- The Reagents & Tools TABLE needs to be removed from the ms file and uploaded separately.

The table is removed from the Ms and will be uploaded separately.

- Materials and Methods should be just Methods

We renamed the heading.

- Our routine image analysis of to be accepted ms detected several figure panel reuses: between Figure 1 and Figure EV1, between Figure 1 and Figure EV2, between Figure 4 and Figure EV4, Figure 5 and EV5, Figure 6 and EV6. All these re-uses need to be explained and mentioned in the EV figure legends.

We added a statement in the EV figure legends explaining the re-use of the figure.

Figure Legends - Comments - Please note that the exact p values are not provided in the legends of figures 1B, D, E, G, H, I, K, L; 2A-D; 3A-L, N; 4B, D, F, H, J, K, L; 5H-J; 6A-D, I, K, M, N, O, P; EV1 A, B, C, E, F, G, I, J; 2B, C, D, G, I; EV3 A, EV4 C, E, G, I; EV5 B, D, F; EV6 B, C, D, F. Please provide exact values as reasonable.

We provided exact values for data that are statistically different.

- Please indicate the statistical test used for data analysis in the legend of figure 2M.

As described in the methods, RNAseq data were analyzed by Novogene, which is based on DESeq2 for dataset analysis of replicates. This statement was added to the figure legend 2M.

- "degenerative" or "degeneration" should be deleted form the second sentence in the abstract.

We deleted degeneration.

EMBO press papers are accompanied online by A) a short (1-2 sentences) summary of the findings and their significance, B) 2-3 bullet points highlighting key results and C) a synopsis image that is exactly 550 pixels wide and 200-600 pixels high (the height is variable). The synopsis image should provide a sketch of the major findings, like a graphical abstract. Please note that text needs to be readable at the final size. Please send us this information along with the final manuscript.

We provided short summary, bullet points and a synopsis image.

Referee 1:

All of the strengths noted before remain: the study is a mechanistic tour de force; the major new finding is the identification of a key role for P2RX4/P2RX7 in "BAT degeneration" and HFD/TNZ (potentially also relevant to long-term involution or de-differentiation associated with aging); the experiments are well-designed and executed; and the overall rigor is high. However, several weaknesses raised earlier were unfortunately not adequately addressed, and they diminish the impact of what could otherwise be a high-quality publication:

1. Unclear focus of the manuscript: Roughly three-quarters of the Results section centers on an artificial pharmacological induction of BAT "degeneration" using CL+Eto, largely in UCP1-/- mice. While the rationale for this model is now more clearly stated-that it provides a simple, rapid approach to induce BAT "degeneration" and thus identify potential mechanisms or drug targets-devoting so much space to characterizing the model detracts from the physiologically relevant sections, which appear only toward the end. Importantly, the final section does not include a comprehensive enough characterization of P2X4/7 function in wild-type mice under HFD, thermoneutrality (TNZ), or aging conditions. In my view, the manuscript does not strike an appropriate balance between these two aspects, leading to confusing and weakly justified statements such as: "the pharmacological treatment regimen based on etomoxir and CL caused inflammatory BAT degeneration, mimicking the phenotype of UCP1-deficient mice." I do not believe mimicking the UCP1-deficient phenotype is of broad interest-unless this were intended as a methods paper, which it is not.

Response: Thank you for carefully reading the manuscript. In the revised version sent to EMBO, we provided a more comprehensive characterization in response to HFD and thermoneutrality. While we think that performing ageing experiments would be interesting, we believe that they are beyond the scope of the manuscript and would delay publication by another two years. As the reviewer questioned the statement that the combined Eto/Cl treatment mimics the phenotype of UCP1-deficient mice, we have deleted it. Furthermore, we added a paragraph summarizing the limitations of the study to the end of the discussion section.

2. Mechanism of ATP secretion remains insufficiently addressed: The role of Pannexin 1 in ATP secretion by brown adipocytes is investigated only *in vitro*, in the absence of nerve terminals or other relevant cell types present *in vivo*. This is a crucial limitation: without *in vivo* validation, the conclusions remain only suggestive and cannot establish the relative contributions of pannexin1 in brown adipocytes versus VNUT-mediated ATP release from other cell types, including adipocytes.

Response: We included a statement at the end of the discussion describing the limitation that we did not study pannexin 1 *in vivo*. Furthermore, we acknowledge in the limitation that we did not provide *in vivo* evidence and therefore cannot exclude VNUT-dependent ATP release by adipocytes or other cell types.

3. Unresolved link between myeloid P2X4/P2X7 and brown adipocyte dysfunction: The new and revised data help clarify the impact of myeloid-expressed P2X4/7 on BAT phenotype. However, the mechanism by which this leads to brown adipocyte dysfunction and BAT degeneration remains unclear. This represents a major gap in what is otherwise a strong mechanistic study.

Response: Sterile inflammation is known to cause inflammatory and fibrotic tissue remodeling. We provide strong evidence that P2X4 and P2X7 are critically involved in macrophage activation and

immune cell recruitment. Notably, we provide a mechanism how stressed adipocytes communicate to tissue-resident myeloid cells via a novel paracrine purinergic axis. Further mechanistic insights about the damaging effects are not within the focus of the current study.

4. Overemphasis on "BAT degeneration": While the revised framing around "BAT degeneration" is better supported by the data, it also detracts from the physiological significance of the work. A key unresolved question is whether the effects of CL+Eto are reversible-and if so, whether reversibility occurs within a timeframe and through mechanisms consistent with de-differentiated BAT in obese or TNZ conditions, where BAT can be reactivated by CL, ATP γ S, cold exposure, or other stimuli.

Response: This study shows that P2X4/P2X7 signaling in macrophages is essential for BAT degeneration. We are surprised that this reviewer, who is questioning the relevance of the pharmacological model, is asking for more experiments based on Eto/CL treatment. Nevertheless, we agree that it would be interesting to study whether the degenerative process is reversible. However, there is no indication that a regeneration process is mediated by purinergic signaling, which is the focus of the current manuscript. If the process is reversible, it is much more likely to depend on the differentiation of thermogenic precursor cells.

5. The role of additional ATP-activated purinergic receptors cannot be disregarded merely because their expression levels appear low in the current assays. Given the opposing roles of P2X5 versus P2X4/7 in BAT function, it is likely that interactions among these receptors-and potentially other purinergic receptors-play an important in vivo role that warrants consideration.

Response: Macrophage-specific P2X4-deficient mice and respective control mice were treated with the antagonistic P2X7 nanobody. Importantly, the treatment with antagonistic P2X7 nanobody alone, has no inhibitory effect on BAT degeneration. Moreover, only the macrophage-specific deletion of P2X4 in combination with P2X7 inhibition protected against loss of UCP1 expression, inflammation, macrophage infiltration, fibrosis and lipid accumulation in BAT as well as glucose intolerance in various models of BAT degeneration. The only possible interpretation of these data is that the combined blockade of P2X4 and P2X7 exclusively in myeloid cells prevents BAT degeneration. In the revised version, we acknowledge P2X5 and its role in adipocyte differentiation and thermogenic function. However, we do not have any evidence that other purinergic receptors are critically involved in the BAT degenerative process.

6. Relying solely on transcriptomics is inadequate to exclude the possibility that CL/Eto exerts BAT-independent effects that indirectly impact BAT function and degeneration.

Response: We disagree. In mice, CL is known to activate cAMP signaling exclusively in white and brown adipose tissues. Although Eto could potentially influence all cells in the body, we did not observe any degeneration in the BAT of mice treated with Eto alone. Therefore, only the combination of Eto and CL induced degeneration of BAT, suggesting that next to brown adipocytes only adipocytes present in WAT could be indirectly involved in this process. Nevertheless, it is highly unlikely that ATP released by white adipocytes would be detected by P2X4/P2X7 receptors expressed by the myeloid cells present in BAT. Moreover, if P2X4/P2X7 were activated in WAT, we would definitely expect to see an effect on gene expression in WAT. This is not the case, and accordingly we believe that there is no need to perform additional experiments using the Eto/CL model.

Referee 2:

In this revised manuscript, the authors have provided additional data to address the reviewers' concerns, along with further clarifications and arguments regarding aspects of the methodology and data interpretation. Overall, the revision strengthens and clarifies the manuscript; however, a few minor updates are still advisable:

1. The authors responded to the shared concern regarding the rationale for focusing on macrophage-adipocyte crosstalk by emphasizing that analyses across several selected immune cell types were performed. While this is a reasonable argument, the approach remains somewhat biased and incomplete. In the current era of single-cell analyses, unbiased approaches that examine all cellular components within adipose tissue and systematically map intercellular communications are both technically feasible and methodologically standardized. It is therefore advisable that the authors acknowledge this limitation in their approach.

Response: Thank you for the advice. We have added a paragraph to the end of the discussion, outlining the limitations identified by the reviewer.

2. The physiological relevance of the Eto/CL model remains debatable. Nevertheless, the authors' arguments highlighting the similarity and relevance of this model to physiological settings such as thermoneutrality are appreciated. It would, however, be advisable for the wording in the manuscript to explicitly acknowledge the limitations of this model.

Response: See point 1.

3. Most importantly, the lack of cell type-specific knockout models remains a significant limitation of the current study. The authors argue that the roles of P2X4 and P2X7 in the myeloid lineage have been validated, citing their restricted expression patterns based on scRNA-seq data, and propose that combining a P2X4 myeloid-specific knockout with a P2X7 antagonist could approximate a myeloid-specific double knockout. However, the absence of detectable expression in other cell types based on scRNA-seq data requires cautious interpretation, given that scRNA-seq often has low sequencing depth and may underestimate low-abundance transcripts, including receptor genes. It is therefore advisable that the authors acknowledge the methodological limitations of their genetic model

Response: See point 1.

Prof. Joerg Heeren
University Medical Center Hamburg-Eppendorf
Biochemistry and Molecular Cell Biology
Martinistrasse 52
Hamburg 20246
Germany

Dear Prof. Heeren,

I am very pleased to accept your manuscript for publication in the next available issue of EMBO reports. Thank you for your contribution to our journal.

Yours sincerely,
